# Analysing Historical and Modelling Future Soil Temperature at Kuujjuaq, Quebec (Canada): Implications on Aviation Infrastructure

Andrew C. W. Leung [1,2,*], William A. Gough [1] and Tanzina Mohsin [1]

1   Department of Physical & Environmental Sciences, University of Toronto Scarborough,
    Toronto, ON M1C 1A4, Canada; william.gough@utoronto.ca (W.A.G.); Tanzina.mohsin@utoronto.ca (T.M.)
2   Data & Services Section, Atmospheric Monitoring and Data Services, Meteorological Services of Canada,
    Environment and Climate Change Canada, Toronto, ON M3H 5T4, Canada
*   Correspondence: andrewc.leung@mail.utoronto.ca

**Abstract:** The impact of climate change on soil temperatures at Kuujjuaq, Quebec in northern Canada is assessed. First, long-term historical soil temperature records (1967–1995) are statistically analyzed to provide a climatological baseline for soils at 5 to 150 cm depths. Next, the nature of the relationship between atmospheric variables and soil temperature are determined using a statistical downscaling model (SDSM) and National Centers for Environmental Prediction (NCEP), a climatological data set. SDSM was found to replicate historic soil temperatures well and used to project soil temperatures for the remainder of the century using climate model output Canadian Second Generation Earth System Model (CanESM2). Three Representative Concentration Pathway scenarios (RCP 2.6, 4.5 and 8.5) were used from the Intergovernmental Panel on Climate Change (IPCC) Fifth Assessment Report (AR5). This study found that the soil temperature at this location may warm at 0.9 to 1.2 °C per decade at various depths. Annual soil temperatures at all depths are projected to rise to above 0 °C for the 1997–2026 period for all climate scenarios. The melting soil poses a hazard to the airport infrastructure and will require adaptation measures.

**Keywords:** time series analysis; climate projection; statistical downscaling; climate change impacts; critical infrastructure vulnerability; Northern Canada; subarctic

## 1. Introduction

The Arctic experiences one of the fastest-warming climates on Earth [1]. In Canada, many indigenous peoples live in remote Northern communities without year-round road access. These communities rely heavily on air transportation to fly in goods, produce and passengers. Most soil temperature measurement stations in Canada, especially in the north, showed a warming trend [2]. Subsurface warming could damage airport infrastructures such as runways, control towers, terminal buildings and fuel tanks, especially for airports underlain by permafrost. Kuujjuaq, formerly known as Fort Chimo, in northern Quebec was chosen as the study area because it has long-term soil temperature data (1967–1995) and it was located on the airport property. Kuujjuaq lacks road access to other communities but is a major transfer hub to other communities in the region, highlighting the importance of air service at this airport. It is also an entry-exit point for cruise ship passengers heading to the Northwest Passage [3]. The permafrost extent at Kuujjuaq has been identified as discontinuous during the study period [4] and this suggests that the airport infrastructure may be vulnerable to melting permafrost [5]. In addition to these pragmatic concerns, Oelke and Zhang [6] describe soil temperature measurements as an important and sensitive climate indicator because this variable reflects the integrated impact of climate processes such as surface air temperature, snowfall, evaporation rate and soil moisture variation [7]. Soil temperature was identified as one of the key drivers for the vegetation phenology and productivity in the Arctic tundra biome [8].

An overall warming trend in air temperature had been observed in this region [9]. However, there is a general lack of subsurface temperature measurements in Canada's north, resulting in observation data gaps for analysis [10,11]. Therefore, subsurface temperature analysis has relied on related proxy variables. For example, one study used Stefan's equation and mean air temperature to estimate subsurface temperatures and permafrost conditions through freezing degree-days in the present and future emission scenarios in northern Canada [12]. Stefan's equation was used in many studies as it required little site-specific information and few parameters input into the equation, but it was ill-equipped to handle multi-layered soil without additional adaptation [13]. A different study coupled water and heat transport into a dynamic global vegetation model to project soil temperatures in the circumpolar region [14]. Another study used the relationship between mean annual air temperature and the existence of permafrost to map the extent of permafrost in Iceland and Nordic countries [15]. Gough and Leung [16] examined permafrost in the Hudson Bay region using a frost number analysis that linked surface air temperature to permafrost conditions and concluded soil moisture played a critical role in determining thermal thresholds. The coupling of air and soil temperatures appeared to be rather site-specific within Canada [17].

One study did use soil temperature records to directly measure the observed soil temperature changes, but it did not conduct future soil temperature projections [2]. This study showed that most soil temperature measuring sites in Canada were located in southern regions. They were often measured at Agriculture and Agri-Food Canada and Reference Climatological Stations (RCS). A soil temperature projection study was conducted in three sites in southern Quebec [18]. Using the Intergovernmental Panel on Climate Change (IPCC) Special Report on Emissions Scenarios (SRES) A2 scenario, Houle et al. [18] projected that the soil would warm by 1.1 °C to 1.9 °C by 2050 and 1.9 °C to 3.3 °C by 2080 depending on the soil depth. In their study, higher temperature increases were observed in summer and there was no clear spatial trend among the three sites. While such borehole measuring sites do exist in more northern regions of Canada, their measuring durations were often too short to perform climatological analysis. From Environment and Climate Change Canada's climate archives, Kuujjuaq was the only site with long-term soil temperature records near the Hudson Bay region suitable for climate analysis. Baker Lake, Nunavut and Schefferville, Newfoundland and Labrador were the other two sites in the region with soil temperature measurements. They had short observation records at 16 years and 3 years respectively. Other long-term soil measuring locations were further south in the Quebec region, further west at Fort Smith in Northwest Territories or in the far north at Clyde River and Resolute in Nunavut.

For this study, the research objectives for this work are to explore the temporal trends of soil temperatures in the Kuujjuaq historical record and to assess the impact of climate change on soil temperatures over the next century. Specifically, the following goals are addressed:

(1) To assess temporal trends of the historical soil temperature record of Kuujjuaq, Quebec and their relationship to concurrent air temperature;
(2) Deploy a novel approach by using statistical downscaling to develop a robust relationship between soil temperatures and a range of atmospheric variables;
(3) Using the relationships developed to project soil temperatures at Kuujjuaq from 1997 to 2086.

## 2. Materials and Methods

### 2.1. Site Characteristics

Kuujjuaq, Quebec (Figure 1) is located at the western shore of the Koksoak River, which flows north and empties into Ungava Bay. The airport is located southwest of the community. There was no vegetation on top of the soil temperature measuring site when the earth thermistors were installed (Figure 2). Adjacent to the soil temperature site were other weather instruments such as an anemometer and a Stevenson screen, and two-

storey buildings such as the radio office and hydrogen building for radiosonde launches. According to the Figure 2 photos, there were shrubs and stunted trees growing around the weather station. During the installation of the earth thermistors at this site, a weather station inspection report on 9 October 1966 stated that "at five feet deep, solid rock was hit and ten-foot level could not be reached, therefore no data will be available for the 300 cm level" [19]. From a weather station inspection report in May 1972, the inspector described the surrounding areas as being characterized by gneiss rocks and the subsurface soil was frozen in permafrost with an active layer thawed at the surface [20]. The report also mentioned that the site was surrounded by lichen woodland forest with tamarack, black spruce, willow and showy mountain ash trees. Vegetation transitioned into tundra about 30 km north of the station.

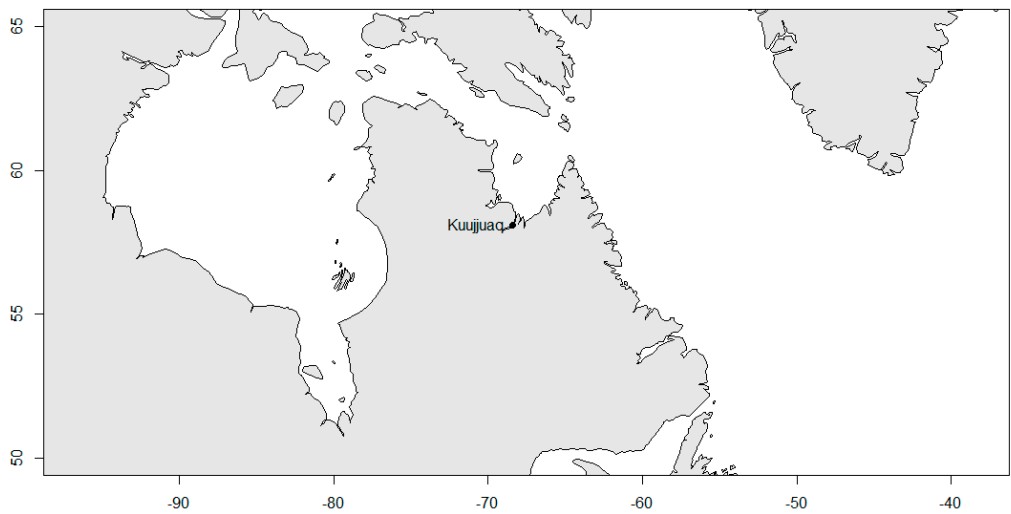

**Figure 1.** Site location: Kuujjuaq, Quebec (58°05′42″ N, 68°25′20″ W).

### 2.2. Data Collection

Soil temperature records at Kuujjuaq were obtained from Environment and Climate Change Canada's Climate Data Online (CDO; https://climate.weather.gc.ca/) (accessed on 25 May 2021). Daily data were available from January 1967 to August 1995 at 5, 10, 20, 50, 100 and 150 cm depths. According to an Environment Canada's undated weather station report, the soil measurement program ceased at this location in September 1995 when the weather station service was contracted out. It was believed that this decision was made as part of the "program review" by the federal government in 1994 to 1995 to lower the expenditure of each department in an attempt to balance the budget [21]. Soil temperatures were measured by Northern Electric Type 14B potted thermistor probes buried at those depths and had a corrected accuracy of 0.1 °C [22]. While soil temperatures were recorded twice daily, this study only used the morning (8:00 a.m. local time) soil temperature measurements due to less missing data in the record [2] and because afternoon (4:00 p.m. local time) measurements were not taken at 50, 100 and 150 cm depths [23].

Surface air temperature at 2 m from the same time period was also obtained from CDO. The daily mean air temperature was derived from the average of highest and low hourly temperatures of each day, from 06:01 Universal Time Coordinated (UTC) to 06:00 UTC the next day. Hourly and daily air temperatures continue to be available to the present day.

(a)

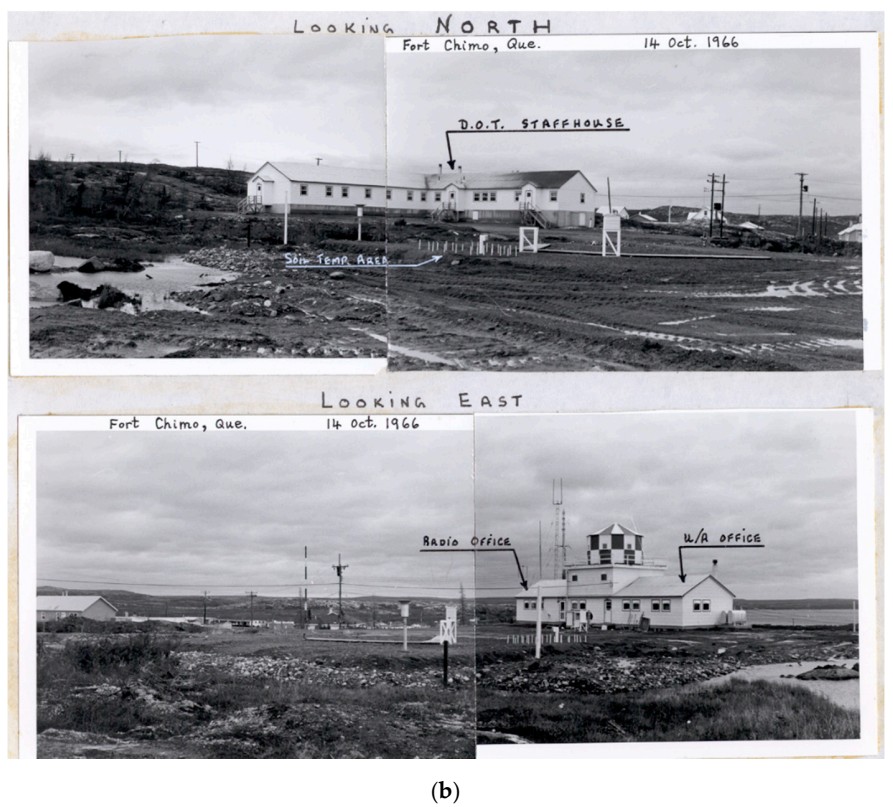

(b)

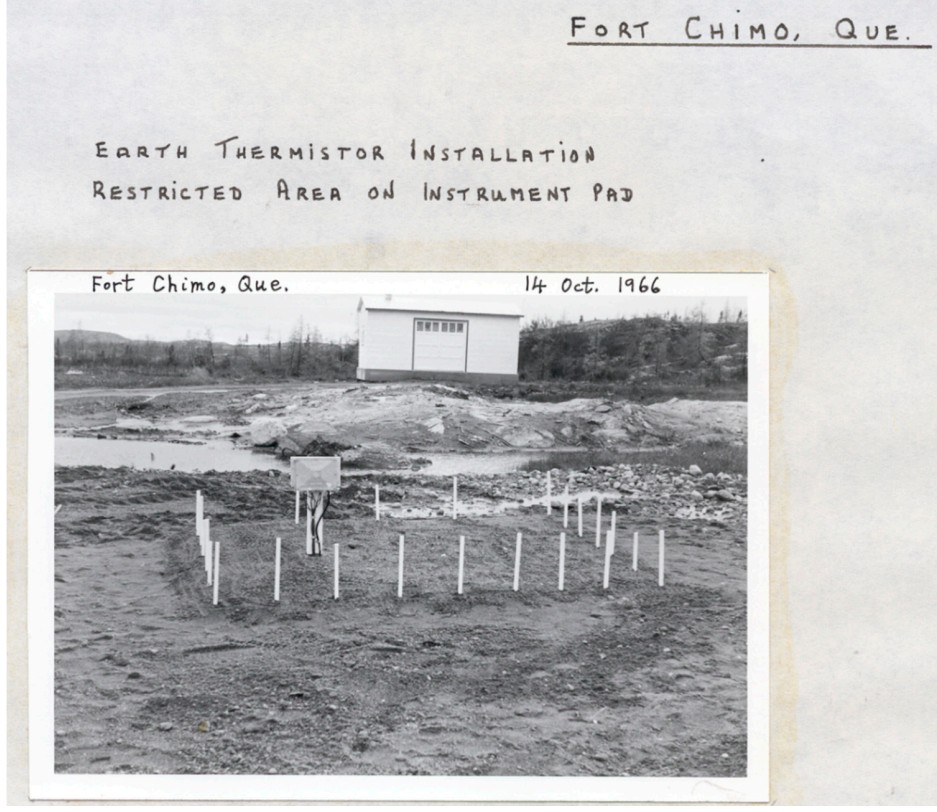

**Figure 2.** Photographs taken during the installation of earth thermistors by Environment Canada's staff at Kuujjuaq (formerly known as Fort Chimo) on 14 October 1966 [19]. The pictures depict (**a**) the surrounding environment and the buildings of the weather station; (**b**) the location and ground conditions where the soil temperature instruments were installed.

### 2.3. Historical Data Analysis

To address the first research goal, historical soil temperature trends were averaged into annual and winter time series for all six levels. Winter was defined as spanning from November to April and summer as May to October given the northern location of this site. Trend analysis was done with the Mann–Kendall test [24,25] and the rates of changes were determined by the Theil–Sen slope estimators [26,27]. Any significant *p*-values influenced by autocorrelated data were adjusted by modifying the variance of the data (Hamed and Rao, 1998). In this modified Mann–Kendall test, a correction factor was calculated to determine the effective number of observations to account for the effects of autocorrelation on the significance level. The variance in a time series dataset with negative autocorrelation was reduced while variance in positive autocorrelation was increased [28]. Finally, the soil temperatures are correlated as a time series to the locally collected air temperature using a Pearson correlation analysis (r) [29].

### 2.4. Statistical Downscaling

To address the second research goal, the Statistical Downscaling Model—Decision Centric version 5.2 (SDSM) software, a tool created by Wilby et al. [30] for modelling surface weather conditions such as temperature and precipitation was used. The software provided tools to determine which atmospheric variables from National Centers for Environmental Prediction/National Center for Atmospheric Research (NCEP/NCAR) reanalysis were best correlated to a predictand. The first step in the use of SDSM was to determine if the observed data (soil temperatures at various depths) can be accurately reproduced by a combination of atmospheric variables taken from the NCEP/NCAR reanalysis gridded data [31]. NCEP/NCAR reanalysis data uses surface, satellite and other data to generate a representative value for the average air temperature change within the locations of the grid based on past data. Since Kuujjuaq airport has the longest observed dataset of all airports in its vicinity, NCEP/NCAR reanalysis data are strongly linked to Kuujjuaq's historical air temperature records. Kuujjuaq's soil temperature data was split into two halves. The first half (1967 to 1980) was used for calibration and the second half (1981 to 1995) was used for verification. The calibration process involves selecting soil temperature at each depth in Kuujjuaq and compared with the NCEP/NCAR reanalysis variables to determine how well those variables were linked to soil temperature with r correlation [29]. The top four variables with the highest r correlation were chosen as the components of the model for the validation process. The weather generator in the SDSM software creates the soil temperature model by using the selected four variables for the verification period. In the verification stage, the model's soil temperature was compared with the observed soil temperature in the same period. Once the verification was satisfactory, the selected variables and the full period of historical soil temperature data (1967 to 1995) were used in the software's weather generator component to project future conditions in conjunction with climate model output.

The modelling efficiency formula (MEF; Equation (1)) evaluates the performance of the SDSM model output of daily soil temperature data based on selected variables and compare that to the observed average [32]. MEF generates a value between $-1$ and $+1$. A positive MEF value closer to $+1$ indicates that the modelled values were a close match with the observed values. A value of zero indicates that the model's predicted value was no better than the observed average value. A negative value indicates the observed average value was better than the model's individual predicted values. Other common model evaluation methods: root mean square error (RMSE; Equation (2)) and mean absolute error (MAE; Equation (3)). A lower RMSE and MAE value would indicate a more accurate model because the predicted values were closed to the observed values (and vice versa). This validation exercise is repeated for all depths.

$$\text{MEF} = \frac{\sum_{i=1}^{n}\left(O_i - \overline{O}\right)^2 - \sum_{i=1}^{n}\left(P_i - O_i\right)^2}{\sum_{i=1}^{n}\left(O_i - \overline{O}\right)^2} \qquad (1)$$

$$\text{RMSE} = \sqrt{\frac{\sum_{i=1}^{n}(P_i - O_i)^2}{n}} \tag{2}$$

$$\text{MAE} = \frac{\sum_{i=1}^{n}|P_i - O_i|}{n} \tag{3}$$

where $n$ = number of observations; $O_i$ = observed value; $\overline{O}$ = mean of observed values; $P_i$ = predicted value

### 2.5. Climate Projections

The climate projections were generated using the relationships between soil temperatures and atmospheric variables in the previous section. However, the model would be projecting air temperature and not soil temperature in the SDSM software. Therefore, it is necessary to establish the relationship between soil temperature and atmospheric variables from the previous section and use that relationship to evaluate how future soil temperature will change due to warmer air temperature.

The climate model chosen to provide the atmospheric variables was the second-generation Canadian Earth System Model (CanESM2) from Climate Model Intercomparison Project Phase 5 (CMIP5). CanESM2 is a model developed by Environment and Climate Change Canada's Canadian Centre or Climate Modelling and Analysis (CCCma) for IPCC Fifth Assessment Report (AR5). It has a horizontal longitude resolution of 2.8125° and latitude resolution of roughly 2.8125°. It captures the observed temperature variability in the 20th century in the Arctic environment reasonably well [33]. In addition, CanESM2 uses the Canadian Land Surface Scheme (CLASS) as its land-surface model, which contains soil temperature input [34], and the CanESM2 model output includes soil temperature as one of the variables [35]. The regional climate model driven by CanESM2, the Canadian Regional Climate Model (CanRCM4), did not have soil temperature as one of its output variables. According to Chylek et al.'s assessment [33], the CanESM2 model is also an improvement over the previous models developed by CCCma because older models overestimated the Arctic warming rate by two to three times. Furthermore, it was shown that a smaller subset of global climate models (GCMs) were generally adequate to reproduce the mean and spread of air temperature and precipitation compared to larger ensemble models in the Hudson Bay region [36]. It should be noted that CanESM5, the successor to CanESM2, was a CMIP6 model published in 2019 [37]. In CanESM5, CLASS along with Canadian Terrestrial Ecosystem Model (CTEM) formed the land component of CanESM5. The only improvement added to CLASS-CTEM since CanESM2 was the introduction of dynamic wetlands and their methane emissions. Furthermore, the methane emission was strictly diagnostic. As shown in Figure 2, the study location was not a wetland. Therefore, CanESM2 was not expected to have a substantial difference in soil temperature prediction as in CanESM5.

The calculation for the rate of change in the future is divided into two parts. In the first part, the soil temperature at each depth was assumed to change at the same linear rate in the future as in the historic trend. In the latter part, additional changes were calculated by using the Localizer Tool to determine the future projections for the change in air temperature at 2 m in different IPCC AR5's Representative Concentration Pathway (RCP) scenarios over the three projection periods (1997–2026, 2027–2056 and 2057–2086) under CanESM2 model within the GCM grid [38]. Change in air temperature was calculated by determining the difference between the historic and projected future mean air temperature. The air temperature change was converted into a percentage and this percentage was applied to soil temperature. Changes in the future soil temperatures were assumed to be the same as in the historical period for the same depth. The relationships established in the previous section were applied to climate model data (replacing the NCEP data used in the calibration process) for three 30-year time periods, ending in years 2026, 2056 and 2086. This is a common approach for modelling future temperatures and a standard for SDSM projection studies.

Each of the future soil temperature depths was projected using three RCP scenarios (RCP 2.6, RCP 4.5 and RCP 8.5). Created for IPCC AR5, RCPs are greenhouse gas concentration trajectories that allow climate projections without assumptions on population growth or economic development [39]. This allows for a wide range of government policies, decision pathways and fossil fuel usage. RCP superseded previous climate change projection scenarios that were based on IPCC's SRES, in which SRES had specific scenarios for population growth, economic development and fossil fuel usage.

RCP 2.6 scenario represents a greenhouse gas emission mitigation scenario, with emissions peaking in 2020 and declining rapidly afterward [39]. RCP 4.5 scenario represents greenhouse gas emissions peaking at around 2040 and then decline while RCP 8.5 scenario represents continuous rising global greenhouse gas emissions in the 21st century. The model output by the software generated 30 years of future daily soil temperatures for 20 simulations at each of the six depths (5, 10, 20, 50, 100 and 150 cm). Each simulation is a synthetic, stochastic model to generate a plausible local climate scenario based on the observed data and climate predictor variables specified by the user [30]. Individual simulations were given equal weighting and the soil temperatures for each day for each level were averaged across the 20 simulations to derive an averaged daily soil temperature for each depth. Then, thirty years of averaged daily soil temperature values were averaged to create a single representative value for that 30-year projection period. The steps were repeated for winter projections by using data from November to April, where the air temperature was below 0 °C during these months [40–42], within the annual projection dataset.

Assumptions

A number of assumptions were made in the process of generating future soil temperatures. It was assumed that the predictor variables with the strongest correlation with the historical daily soil temperature would also be strongly correlated in the future. Moreover, while the selected predictor variables would be strongly correlated for the entire study period, other predictor variables could be better correlated in winter or particular months. Aside from assuming the historical warming rate would form as the baseline trend, the changes to air temperature under the three RCP scenarios were assumed to have the same percentage of effects on the soil at each depth. The variability of the soil temperature at each depth in the future was assumed to be identical to the historical level. Snow depth in winter and any vegetation such as grass and shrubs growing on top of the soil were ignored even though both factors influenced the radiative forcing by reducing the incoming solar radiation that reached the surface of the soil. Snow depth in the future was also assumed to be held constant even though the depth was decreasing during the historical period at this location [40–42]. The r correlation would not consider the thermal lag, particularly at deeper depths. The temperature in deeper soils had less magnitude and was more distant from the fluctuations in air temperature. Moisture level, soil type, organic matter contents and distribution within the soil were assumed to remain constant in the future even though these variables could change over time. The study also speculated that the instrument remained accurate and that regular maintenance on the instrument was performed without disturbance to the soil.

## 3. Results

### 3.1. Historical Analysis

The statistical analyses for the period of 1967 to 1995 for the six soil temperature depths are reported in Table 1. The analyses include average annual and average temperatures, variability and temporal trends. The temporal trends were done using the Mann–Kendall test. Theil–Sen slope estimators determined the magnitudes of the rates of changes. The significance, *p*-values, was corrected for autocorrelation as appropriate. As noted above "winter" was defined as November through April. Table 2 reports a similar analysis for the concurrent air temperatures at 2 m measured at the same location.

**Table 1.** Historic average soil temperature and the rate of change over time for soil temperature at depths from 5 to 150 cm from 1967 to 1995. Bolded numbers indicate trends that are significant at *p*-values less than 0.10 (^), 0.05 (*) and 0.01 (**).

| Soil Depth | Mean (°C) | | Trend (°C/Decade) | |
| --- | --- | --- | --- | --- |
| | **Annual** | **Winter** | **Annual** | **Winter** |
| 5 cm | −2.1 | −11.1 | +1.2 * | +1.9 * |
| 10 cm | −2.4 | −10.8 | +1.1 * | +1.7 ^ |
| 20 cm | −1.9 | −9.9 | +1.3 ** | +1.4 ^ |
| 50 cm | −1.4 | −7.9 | +1.0 * | +1.2 |
| 100 cm | −1.6 | −6.2 | +0.9 ^ | +0.8 |
| 150 cm | −1.9 | −4.2 | +1.1 ** | +1.0 |

**Table 2.** Pearson r correlation (r) between air temperature at 2 m and soil temperatures.

| Soil Depth | r Coefficient (Daily) | | r Coefficient (Monthly Average) | |
| --- | --- | --- | --- | --- |
| | **Annual** | **Winter** | **Annual** | **Winter** |
| 5 cm | 0.845 | 0.453 | 0.955 | 0.746 |
| 10 cm | 0.847 | 0.452 | 0.951 | 0.733 |
| 20 cm | 0.848 | 0.450 | 0.946 | 0.722 |
| 50 cm | 0.819 | 0.391 | 0.906 | 0.597 |
| 100 cm | 0.721 | 0.268 | 0.797 | 0.407 |
| 150 cm | 0.455 | 0.087 | 0.514 | 0.147 |

Mean annual soil temperature at 5 cm was −2.1 °C and the temperature was becoming slightly warmer with increasing depth to 50 cm (Table 1). Below this depth, soil temperature became colder. For winter, the soil temperature closest to the surface was the coldest and became warmer with depth. All soil depths, except 100 cm, were warming significantly at $p < 0.05$ on an annual scale and 100 cm depth warmed at a lower significance level, $p < 0.10$. However, only 5 cm was warming significantly at $p < 0.05$ in winter. Both 10 cm and 20 cm warmed at a significance level of $p < 0.10$. Surface soil at 5 cm is warming at about 1.2 °C per decade annually. In winter, the warming rate was faster, at about 1.9 °C per decade at the same level. At all of the depths other than 100 cm and 150 cm, winter's warming rate was larger than the annual rate.

The results of the r correlation between coincident air and soil temperatures are presented in Table 2. The relationship is strongest for the upper soil levels and for annually-averaged data as expected. The transfer energy into and out of the soil is determined by soil type and moisture content. This leads to the muting of the surface forcing and thermal lags. Hence the lowest correlations occur for the daily winter date at the 150 cm depth. However, the upper 50 cm correlates well with the surface forcing and this bodes well for the SDSM modelling in the next section.

### 3.2. Statistical Downscaling

Using SDSM, historical soil temperature data (1967–1995) was statistically modelled using a suite of variables taken from the NCEP data set. The simulations included those created using three, four and six NCEP variables and four CanESM2 variables to test the effects in varying the number of variables used as well as comparing the performance between NCEP and CanESM2. The top four atmospheric variables for all soil depths are surface-specific humidity, mean air temperature at 2 m, 500 hPa geopotential height and specific humidity at 850 hPa. The linkage to the 2 m atmospheric temperature (Table 3) is expected given the results of the previous section. In most cases, the CanESM2 variables had a stronger correlation with the soil temperature than NCEP's corresponding variables. While NCEP's specific humidity at 850 hPa had a stronger correlation with soil temperature than CanESM2's corresponding variable, the difference in correlation between the two models was very small and less than the correlation differences in other variables where

CanESM2 had a stronger correlation. The strength of the relationship between the variables and soil temperature decreases as soil depth increases. Figure 3 shows the results of various SDSM simulations compared to observations for the 5 cm depth. The reproduction of observed temperatures by the SDSM modelling is particularly good for the summer months but the models consistently underrepresent by a few degrees Celsius for the winter months (Figure 3a) suggesting that surface processes such as snow cover and phase change within the soil may not be captured by the model. The monthly variance pattern was also reproduced, but the greatest difference was found in March (Figure 3b). Differences between the number of variables used in the NCEP model or between the NCEP and CanESM2 models were barely distinguishable as seen in Figure 3. It was noted by Hassan and Harun (2012) that there was no standard rule for selecting which predictor variables to use and the process itself was described as difficult and tricky. Since CanESM2's predictor variables were shown to be better correlated to soil temperature than NCEP's in most cases and at most depths, CanESM2 model was chosen to repeat statistical downscale modelling at other depths.

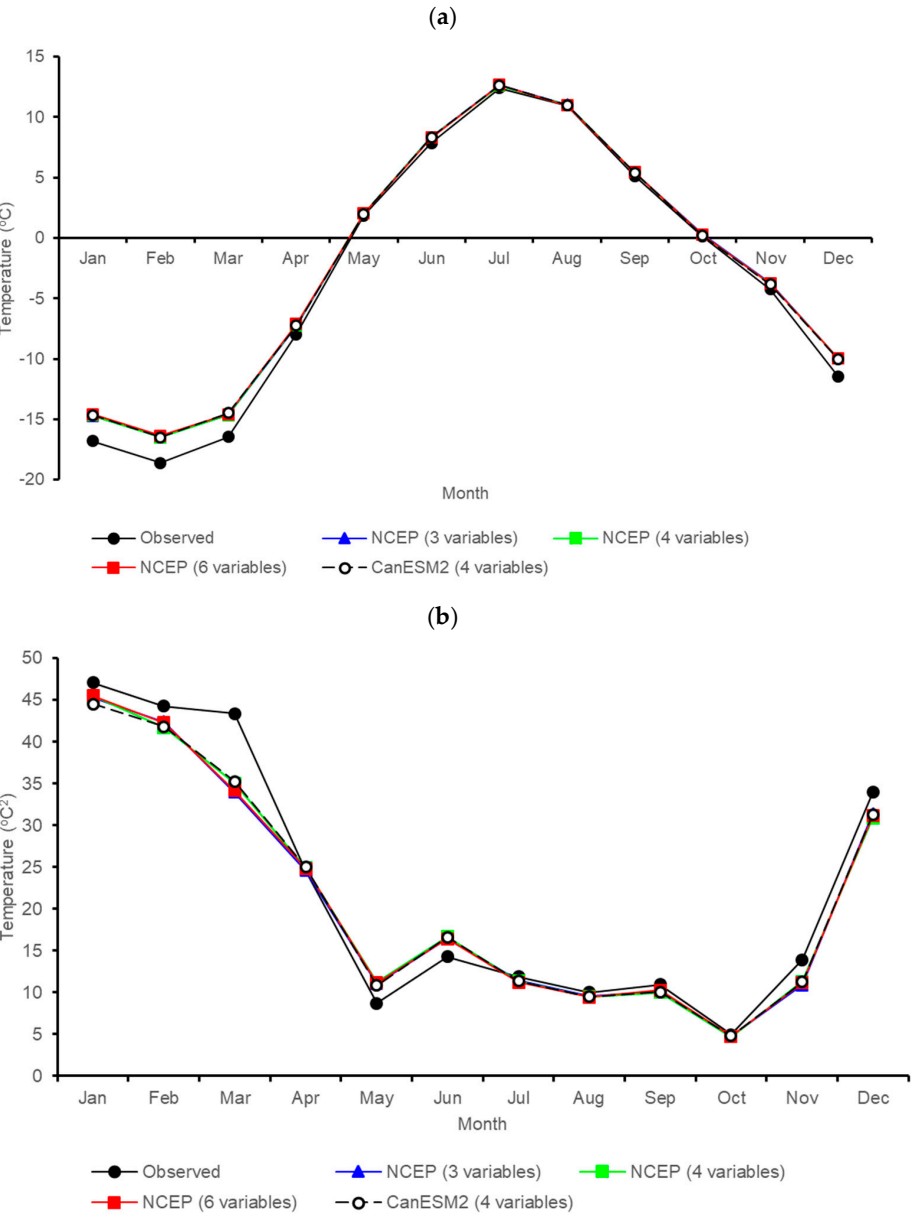

**Figure 3.** Comparison between the observed monthly 5 cm soil temperature and the predicted values from the CanESM2 and NCEP models for: (**a**) mean; (**b**) variance.

**Table 3.** r correlation between the four selected variables and observed soil temperature, with bolded numbers indicating the model variable having a stronger correlation on the soil temperature than the other model.

| Soil Depth | r Correlation | | | | | | | |
|---|---|---|---|---|---|---|---|---|
| | Surface Specific Humidity | | Mean 2 m Air Temperature | | 500 hPa Geopotential Height | | Specific Humidity at 850 hPa | |
| | NCEP | CanESM2 | NCEP | CanESM2 | NCEP | CanESM2 | NCEP | CanESM2 |
| 5 cm | 0.825 | 0.828 | 0.825 | 0.821 | 0.696 | 0.756 | 0.700 | 0.699 |
| 10 cm | 0.826 | 0.832 | 0.828 | 0.830 | 0.693 | 0.752 | 0.700 | 0.697 |
| 20 cm | 0.812 | 0.824 | 0.806 | 0.825 | 0.673 | 0.732 | 0.689 | 0.687 |
| 50 cm | 0.817 | 0.833 | 0.811 | 0.831 | 0.658 | 0.711 | 0.694 | 0.682 |
| 100 cm | 0.728 | 0.758 | 0.722 | 0.763 | 0.557 | 0.601 | 0.618 | 0.599 |
| 150 cm | 0.466 | 0.504 | 0.462 | 0.525 | 0.314 | 0.337 | 0.390 | 0.365 |

Comparisons between observed and CanESM2 modelled soil temperature at 10 to 150 cm depths were shown in Figures 4–8. At 10 cm, observed temperatures were higher than modelled in all the months and the difference between observed and modelled were greatest in winter. The model produced a similar variance in temperature as the observed from April to October. Over-estimating the coldness of soil temperature in the CanESM2 model was also found in all of the months at 20 to 150 cm depths, with winter having the greatest overestimation while summer had the least. In general, the variability of the modelled output was similar to the observed data's variability, which is an indication that the model's variability was able to reproduce the historical variability. The variance in the historical data is mainly attributed to natural variability of the temperature in the soil. The variances of observed 5, 100 and 150 cm were greater than the modelled in winter but 5 cm had less variance in the observed than the modelled in summer. The variance of modelled 150 cm temperature approached $0 \, °C^2$ from June to November but the observed was much higher in July and August. The variances of observed 10 to 50 cm were generally higher in the modelled data than the observed in most of the months.

The fidelity of the modelling to observations was measured using Equations (1)–(3) and the results were reported in Table 4. The performance of the modelling is consistent through to 100 cm and decreases below 100 cm. As all MEF values were above 0, the individual modelled values across all depths were a close match with the observed values and were better than using the observed average value as a representative for their respective depths. Both RMSE and MAE values decreased as depth increased. Since a lower RMSE and MAE value indicates a more accurate model, the model produced better-predicted values as the depth increased.

**Table 4.** Modelling efficiency (MEF), root mean square error (RMSE) and mean absolute error (MAE) calculation for the CanESM2 model output.

| Modelled Depth | MEF Value (−1 to +1) | RMSE | MAE |
|---|---|---|---|
| 5 cm | 0.83 | 4.590301 | 3.471129 |
| 10 cm | 0.83 | 4.185074 | 3.088709 |
| 20 cm | 0.84 | 3.778079 | 2.731346 |
| 50 cm | 0.85 | 3.160122 | 2.140639 |
| 100 cm | 0.81 | 2.825081 | 1.797228 |
| 150 cm | 0.66 | 2.702048 | 1.673864 |

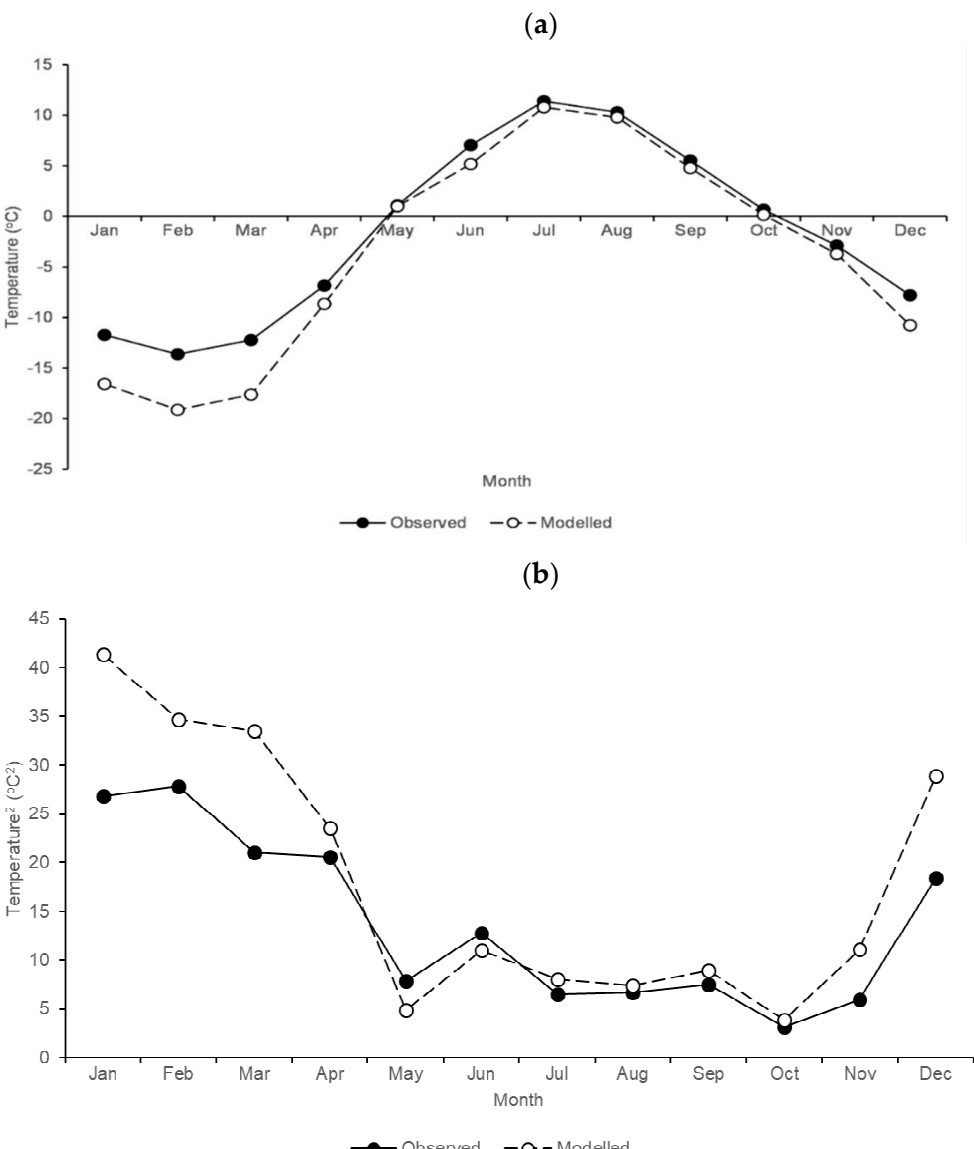

**Figure 4.** Comparison between the observed monthly 10 cm soil temperature and the predicted values from the CanESM2 and NCEP models for: (**a**) mean; (**b**) variance.

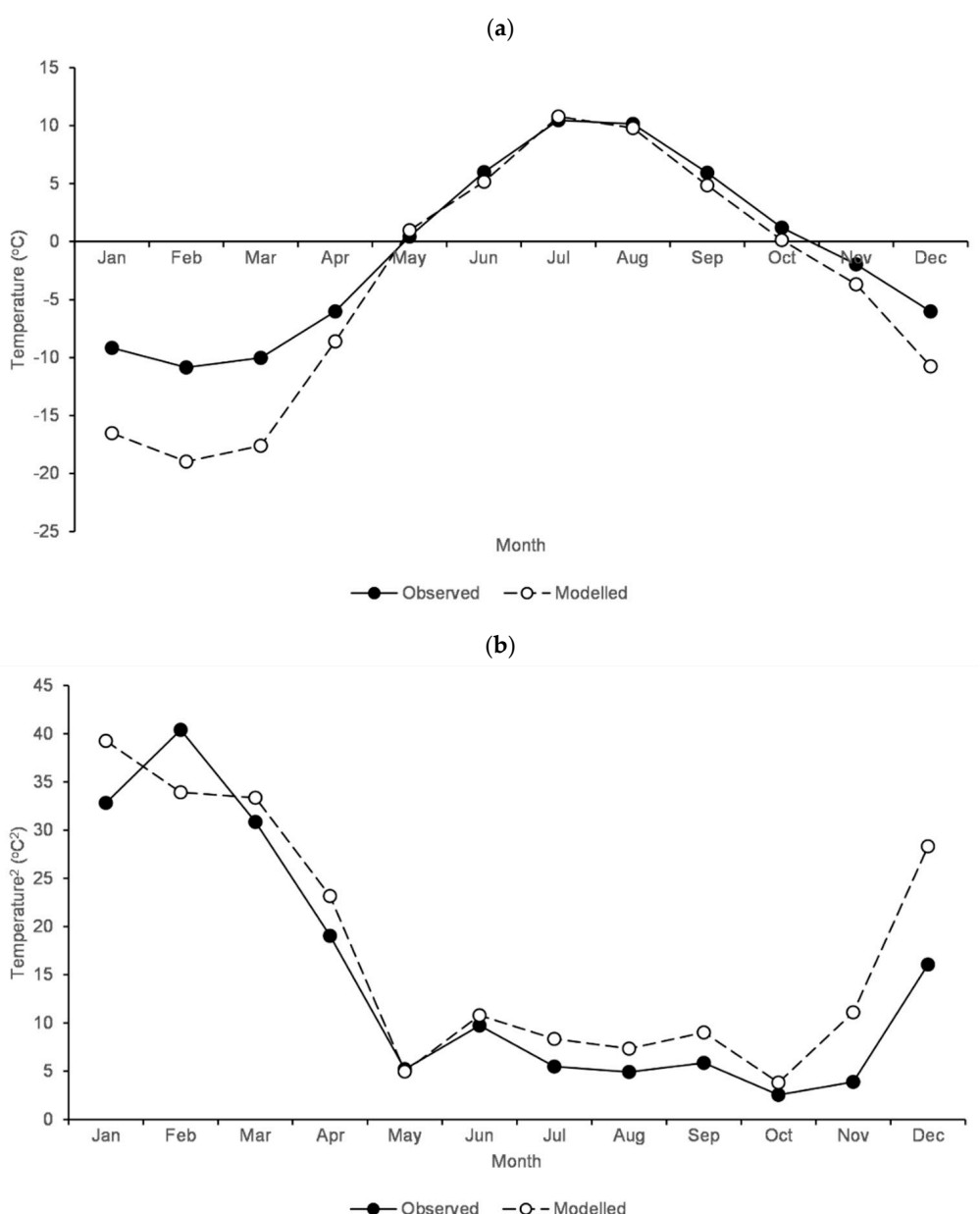

**Figure 5.** Comparison between the observed monthly 20 cm soil temperature and the predicted values from the CanESM2 and NCEP models for: (**a**) mean; (**b**) variance.

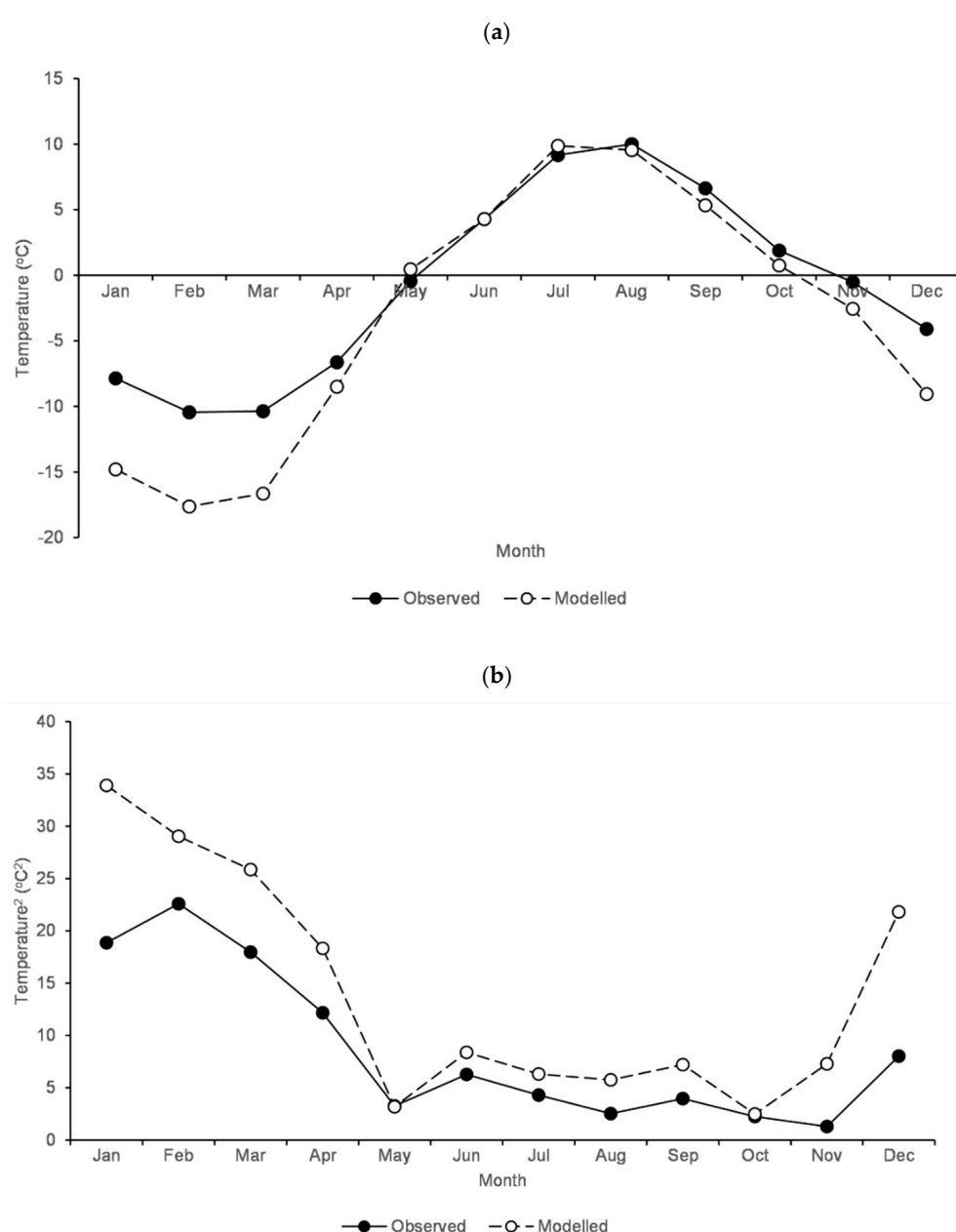

**Figure 6.** Comparison between the observed monthly 50 cm soil temperature and the predicted values from the CanESM2 and NCEP models for: (**a**) mean; (**b**) variance.

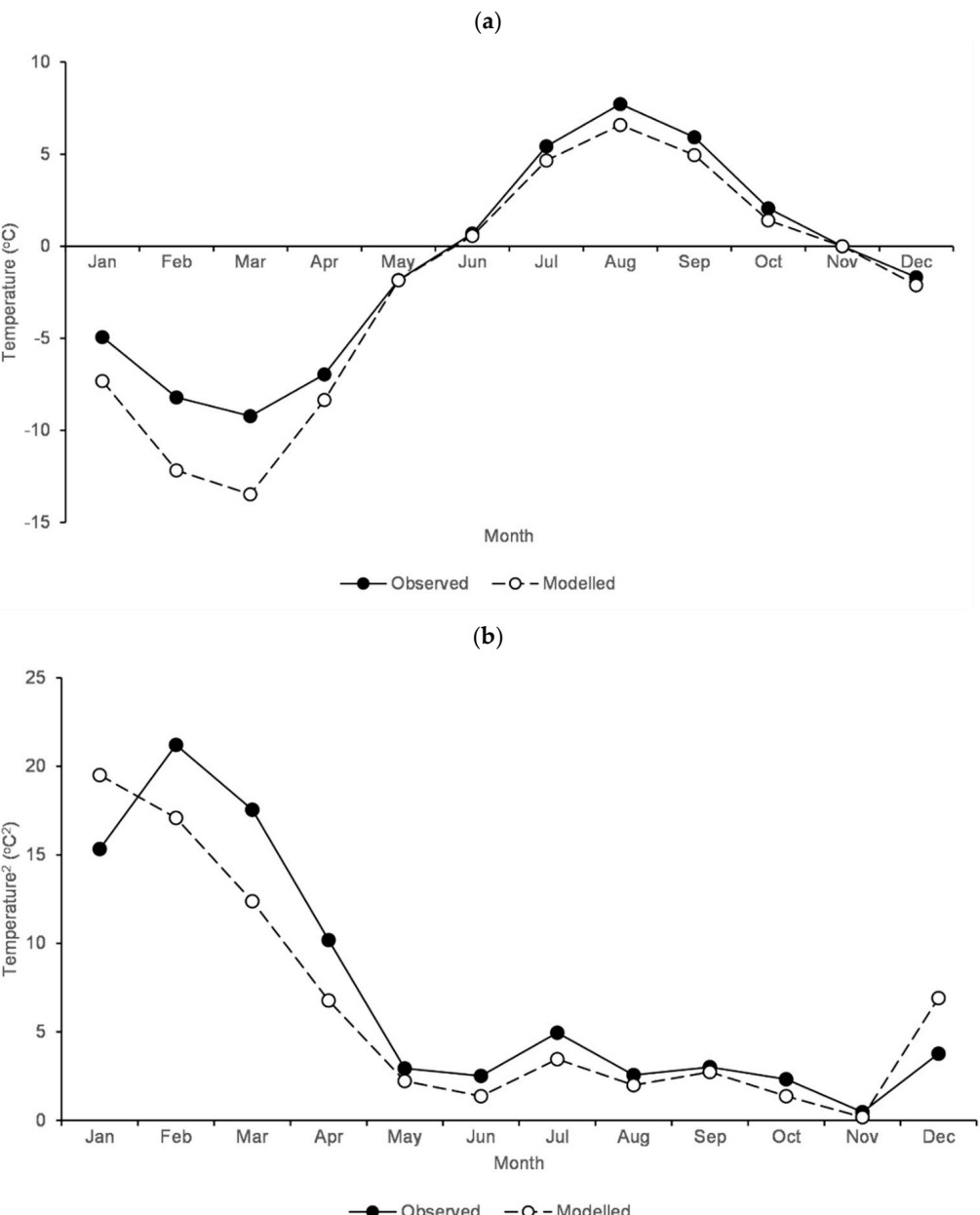

**Figure 7.** Comparison between the observed monthly 100 cm soil temperature and the predicted values from the CanESM2 and NCEP models for: (**a**) mean; (**b**) variance.

**(a)**

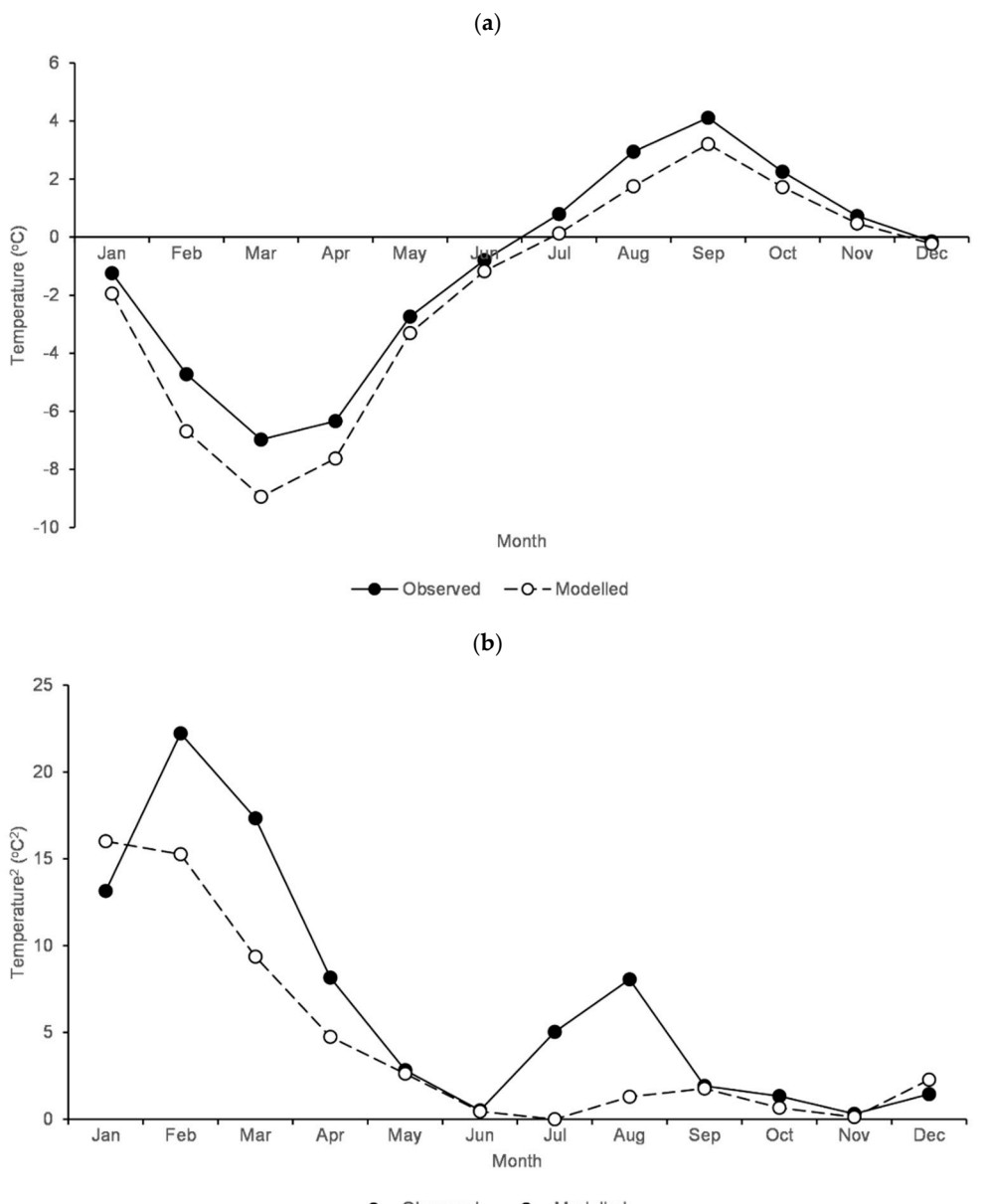

**(b)**

**Figure 8.** Comparison between the observed monthly 150 cm soil temperature and the predicted values from the CanESM2 and NCEP models for: (**a**) mean; (**b**) variance.

*3.3. Projections*

3.3.1. Annual

Future climate projections were conducted at the various depths sampled, first on all 12 months (Figure 9) followed by just winter months (Figure 10). In the projections that used all 12 months, soil temperature exceeded 0 °C in the first projection period (1997 to 2026) in each of the CanESM2 model's RCPs at all depths. During this period, there were few temperature differences between the three RCP trajectories. By the second projection period (2027 to 2056), the projected RCP 8.5 soil temperature increased the greatest and deviated from RCP 2.6 and RCP 4.5 projections. RCP 4.5 projected a slightly warmer soil temperature than RCP 2.6. In the third projection period (2057 to 2086), the warming of RCP 8.5 increased at a greater rate than between the first and second projection periods. The warming rate for RCP 4.5 stayed approximately the same and RCP 2.6 slowed down.

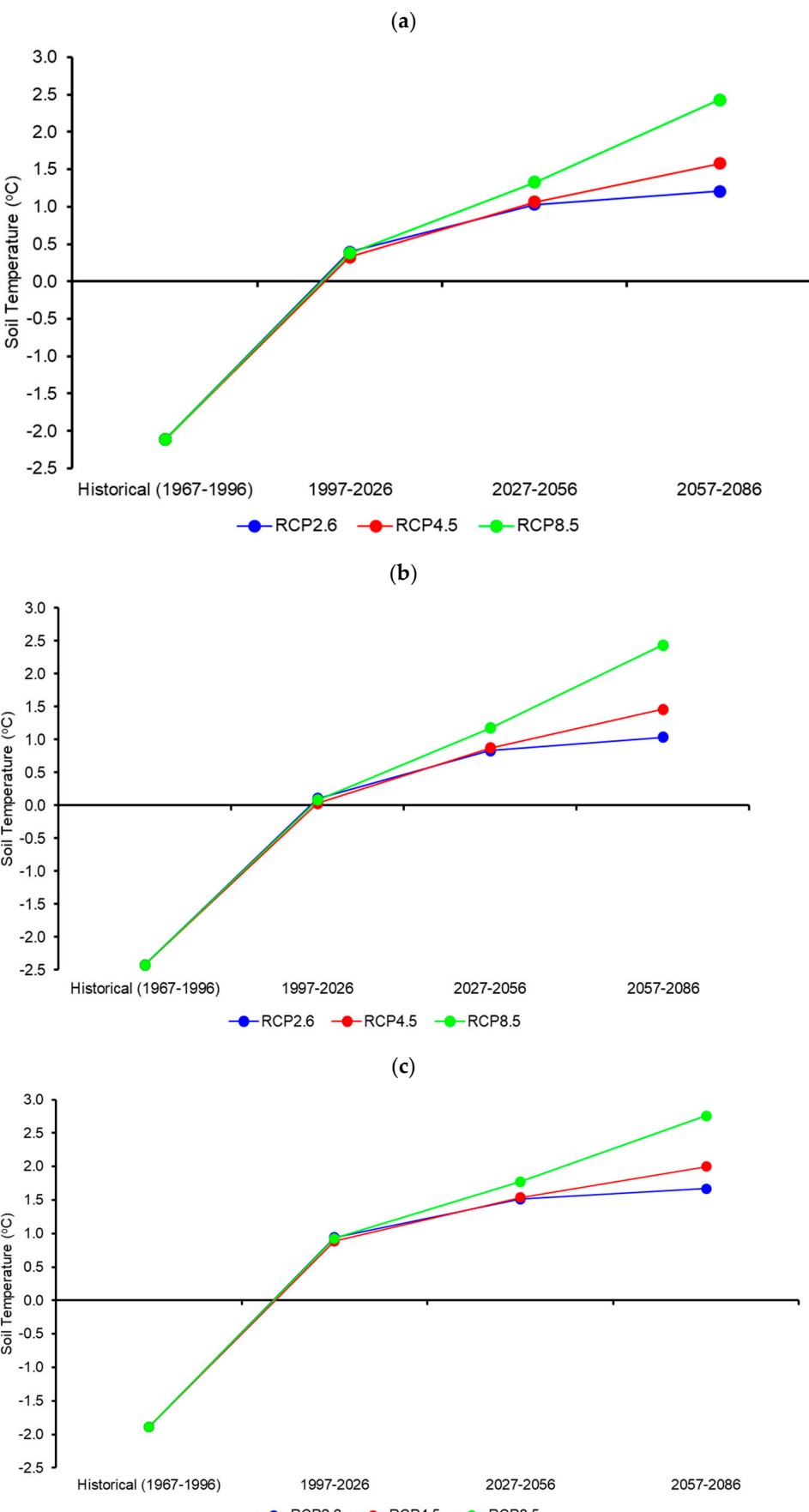

**Figure 9.** *Cont.*

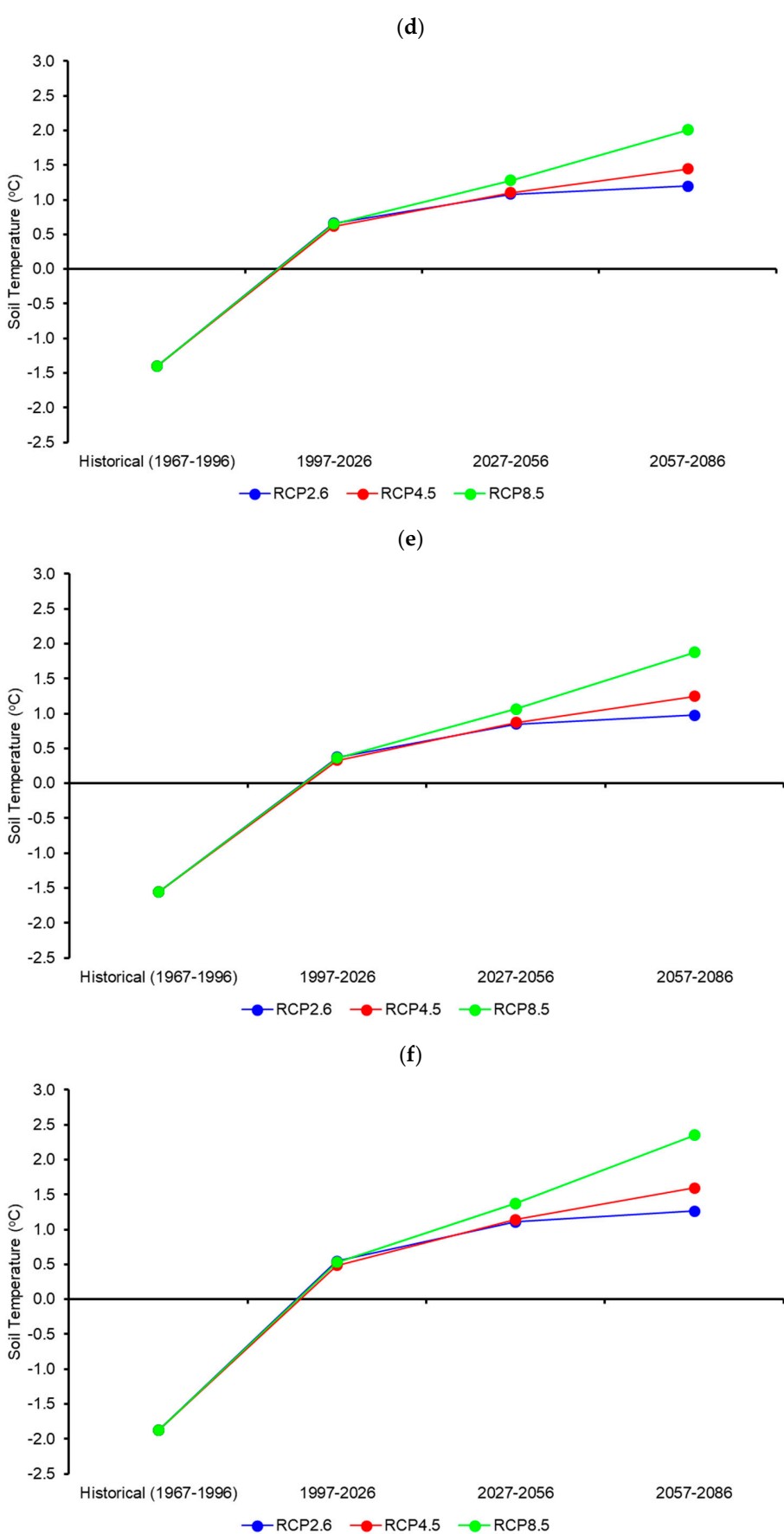

**Figure 9.** Annual soil temperature projections from 1997 to 2086 for depths at: (**a**) 5 cm; (**b**) 10 cm; (**c**) 20 cm; (**d**) 50 cm; (**e**) 100 cm; (**f**) 150 cm.

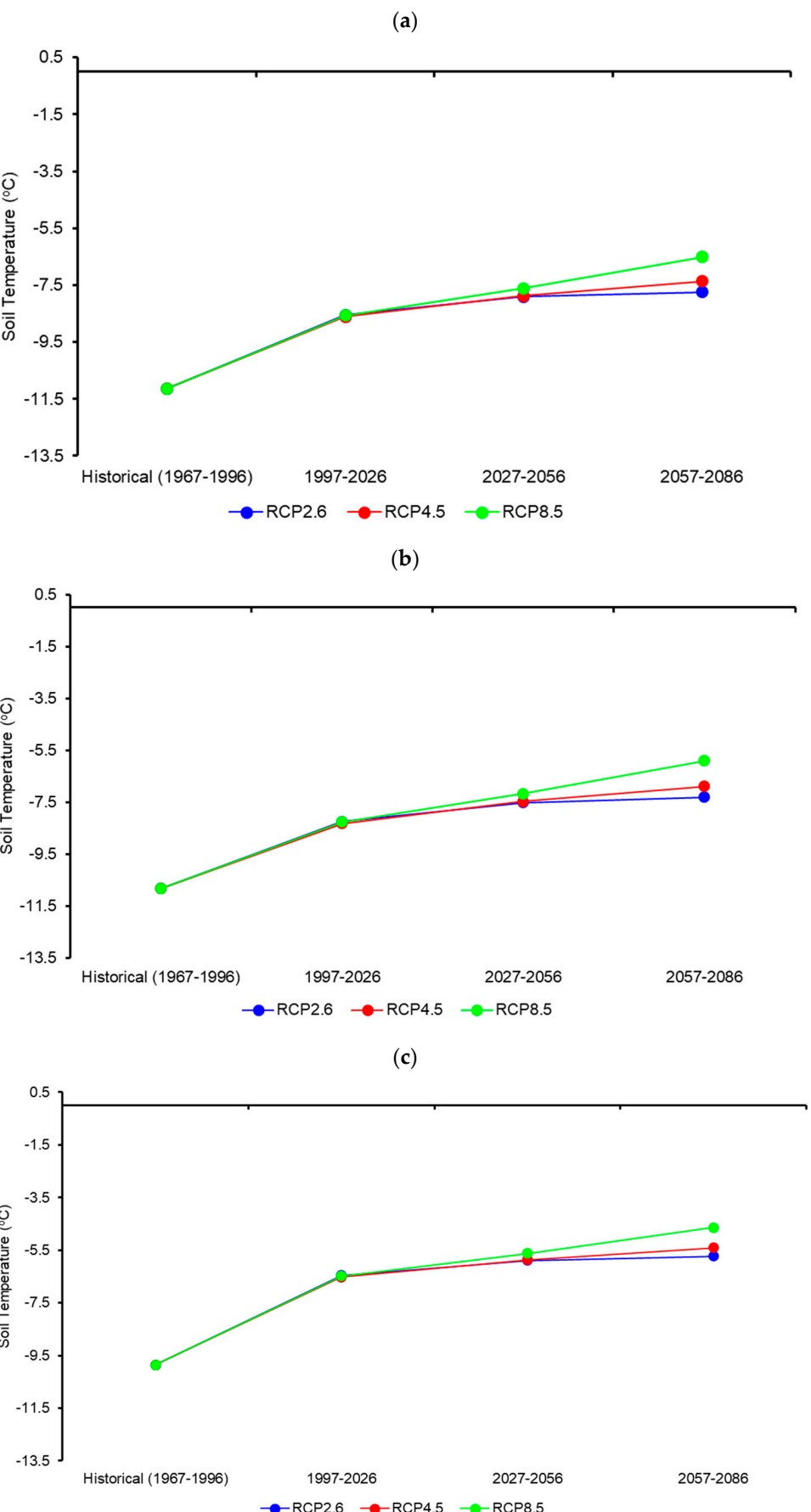

**Figure 10.** *Cont.*

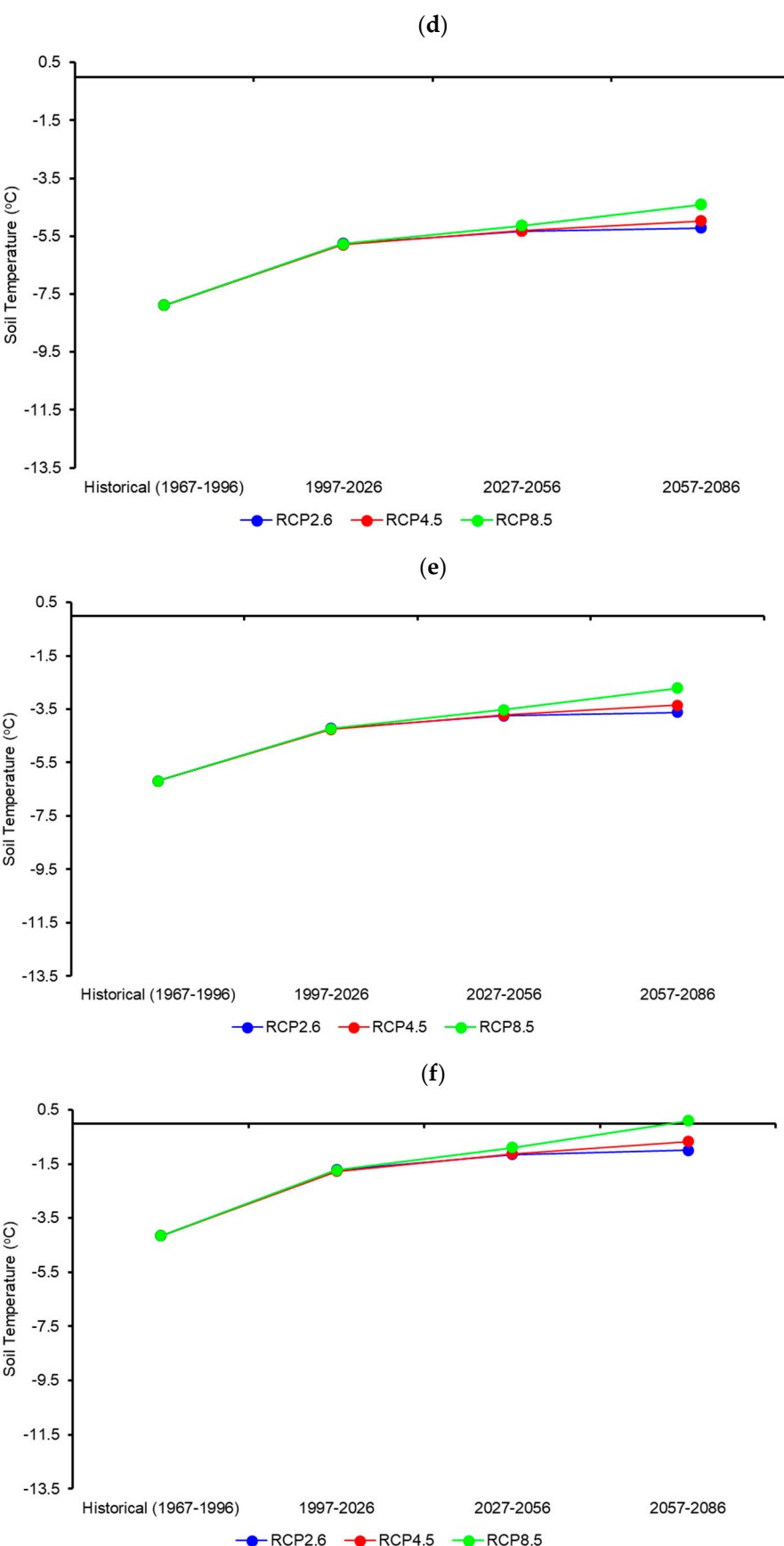

**Figure 10.** Winter soil temperature projections from 1997 to 2086 for depths at: (**a**) 5 cm; (**b**) 10 cm; (**c**) 20 cm; (**d**) 50 cm; (**e**) 100 cm; (**f**) 150 cm.

3.3.2. Winter

In winter months, projections consistent with the local definition of winter, November to April were considered. Winter projections (Figure 10) showed the same patterns as annual projections. Winter soil temperatures were projected to remain below 0 °C at all depths in each scenario with the exception of RCP 8.5 at 150 cm by the third projection period (2057 to 2086). The warming rate for the RCP 8.5 scenario in winter appeared to be at a linear rate while RCP 2.6 and RCP 4.5 scenarios warmed at a slower rate than RCP 8.5 across the three projection periods.

## 4. Discussion

### 4.1. Historical Trends and Model Fitting

In this study, the long-term trends of soil temperature at Kuujjuaq were assessed by using an in-situ measurement record. Between 1967 and 1995, annual soil temperature warmed by 0.87 to 1.32 °C per decade and winter soil temperature warmed by 0.84 to 1.85 °C per decade (Table 1). Soil at 5, 10, 20, 50 and 150 cm depths showed significant warming with $p < 0.05$ and 100 cm depth showed a statistical significance at $p < 0.10$. The soil temperature was strongly positively correlated with air temperature (Table 2). The difference in the correlation between air temperature and soil temperature at various depths were mainly attributed to the thermal capacity of soil that acted as a buffer against radiative forcing from the surface. As the depth increased, the buffer capacity was higher. In addition, time lag was also responsible for different response time to the surface warming or cooling. Interestingly, a study conducted similar soil temperature analysis at Kuujjuaq for a shorter time period (from 1967 to 1989) and found that the increase in soil temperature at all depths did not appear to be associated with the increase in air temperature [43]. That study suggested that environmental conditions other than air temperature were responsible for the increase in mean soil temperature. However, the study also mentioned that short time series analysis might not capture the variability and significant trends and inferences drawn from the results should be treated with caution. An additional six years of soil temperature records in daily and monthly average scales as well as linkage with the GCMs during the model validation process strengthened the validity of this study's claims that there was a strong correlation between soil and air temperatures at all depths. The Pearson correlation coefficients between air and soil temperatures at Kuujjuaq were also similar to other discontinuous permafrost regions in the Arctic [6].

The number of variables beyond four and the use of NCEP vs. CanESM2 did not substantially improve the model accuracy, however, CanESM2 was better correlated to soil temperature than NCEP in most cases (Table 3). It was not unsurprising to see CanESM2 perform better than NCEP since a comparison between CMIP5, CMIP6 and NCEP models in Canada revealed that NCEP overestimated the number of freeze-thaw cycles at the study location in the historical period [44]. While surface air temperature's correlation with soil temperature at 5 cm was established without using model data as illustrated in Table 2, other variables with high correlation in Table 3 could be explained by local weather. Surface-specific humidity is linked to the short term of past weather. If the surface humidity is high, then the soil is likely to be moist and caused by recent precipitation events (rain or snow). Precipitation events are almost always accompanied by clouds which lower the solar radiation reaching the surface and reduce the surface heating. Moreover, some of the energy is used as latent heat of evaporation. Drier soils at the surface allow greater temperature increase even with the same amount of energy. Specific humidity at 850 hPa is also linked to clouds and cloud cover is directly linked to a reduction in solar radiation heating on both the air and soil close to the surface. The strong correlation between soil temperature and 500 hPa geopotential height appeared to be counterintuitive at first, since 500 hPa geopotential height is linked to large-scale mid-troposphere circulation [45] and distant from the surface. Upon closer examination, the synoptic-scale circulation patterns occurring at 500 hPa geopotential height are responsible for air mass and frontal movements which in turn affects mesoscale phenomenon such as high-pressure systems, squall lines

and convective storms. They are all associated with the variation in solar radiation, cloud cover, precipitation, snow cover and air temperature at the surface [43]. Therefore, all of the top four variables are related to surface heating.

The model performed reasonably well as demonstrated by positive MEF values across all depths (Table 4). While MEF decreased somewhat at 150 cm depth, the RMSE and MAE decreased as the soil layer got deeper. This suggested that the model error was getting smaller when the depth increased and likely due to lower temperature variations in deeper soils. The model performed particularly well from April to November in all soil depths (Figures 3a, 4a, 5a, 6a, 7a and 8a). The historical air temperature warming at Kuujjuaq is consistent with temperature warming trends in the Hudson Bay Region [9]. Leung and Gough [9] and this study showed that the largest temperature warming occurred during winter.

Compared with Tam et al.'s study [12], this study provides a more accurate baseline and future temperature for soil because this study used in-situ measurements rather than using air temperature and Stefan's equation to determine the ice thickness level. Stefan's equation requires the thermal conductivity of the soil as an input, which would require a site visit to obtain a soil sample for lab analysis, while this study does not require this variable. Coupling the air and soil temperature also ensured that the relationship between the two variables was directly measured rather than indirectly linked by calculating frost factor from Stefan's equation to determine the permafrost conditions, which could not generate an actual soil temperature value.

### 4.2. Projections

Under projected warming (Figure 9), the annual soil temperature would be above 0 °C in the first projection period (1997–2026). In the second projection period (2027–2056), the warming rate will be similar for RCP 2.6 and RCP 4.5 scenarios while RCP 8.5 would have the greatest warming rate. The warming rate for RCP 8.5 in the third projection period (2057–2086) is further enhanced while RCP 4.5's warming rate will be progressing at approximately the same rate as between the first and second projection periods. RCP 2.6 scenario's warming rate slows down in the third projection period, which is largely explained by a reduction in global greenhouse gas emissions after 2020 [39]. A combination of reduction in fossil fuel consumption, government policy implementations to drastically reduce greenhouse gas emissions, improved efficiencies in existing technologies (e.g., automobiles and heating) and new technologies (e.g., carbon capture and storage) would lead to reduced radiative forcing in the atmosphere and cause lesser warming starting from second projection period. Winter projection trends in different RCP scenarios were similar to the annual's corresponding RCP trends (Figure 9). For RCP 4.5 and RCP 8.5 scenarios, Phillips et al. [46] suggested that the slowdown in the warming rate across different scenarios may be explained by soil drying. They found that under RCP 4.5 and RCP 8.5, 15 CMIP5 models showed soil temperature warming at a rate of 10% less than air temperature. They also hypothesized that soil drying leads to reduced thermal conductivity and heat capacity in the soil. However, after soil was dried, there would be more warming from sensible heat than latent heat as evaporation decreased.

At the 20 cm depth, the Zhang et al. [47] simulations suggested that the soil warming rate was 0.8 °C on a national level yet their model output indicated that the soil actually got cooler (−0.6 °C) at Kuujjuaq during the 20th century. The cooling in this area was cited as an exception to the general warming but shallow layers of soil had increased by up to 2 °C since the mid-1990s [48]. Unfortunately, this study site measurements stopped in 1995 and could not offer evidence to support this claim. This study did find that the 20 cm soil increased significantly ($p < 0.01$) at 1.3 °C per decade during the study period (Table 1). The strong correlation between air and soil temperatures close to the surface was not unexpected since the radiative forcing from the surface contributes to the warming of those soils. Deeper soils are less likely to receive surface energy by heat transfer. Another interesting observation was that the winter soil temperatures at 5 and 10 cm were warming

faster than the annual trend. This showed that the magnitude of warming was greater from winter (November to April) than in summer (May to October) at Kuujjuaq even though about 23 cm of snow cover was present during the winter period and that the snow cover reduced the heat conduction transferred from air to soil. There are two possible explanations to explain this observation. The presence of soil layers could result in a delay in the heat transfer process, leading to a thermal lag. Another possibility was the latent heat effect in the soil that slowed the rising of soil temperature due to phase change, known as the zero-curtain effect [49]. This effect is most commonly observed during late fall/early winter and late winter/early spring in cold climates [50].

This study's projections also provided more information on permafrost degradation rate than Tam et al. [12]. Tam et al. determined the future permafrost conditions based on the relationship of air temperature and frozen ground based on Stefan's equation. This study coupled the air temperature with soil temperature with atmospheric variables for projections. Instead of outputting a ratio that determines the coverage of the permafrost, this study provided a projection for the actual soil temperature.

The observed difference between soil temperature and air temperature in Canada varies greatly and is highly location-dependent, with some sites showing the increase in air temperature was less than that of soil temperature while others showed greater warming in soil temperature than air temperature [47]. At Kuujjuaq, Zhang et al. [47] found that the difference between the changes in soil temperature was greater than air temperature by 0.1 to 0.2 °C between the 1901–1910 and the 1986–1995 periods. Other variables that affect the relationship between air and soil temperatures include thermal conductivity and the type of soil at the site. Wen et al. (2003) studied the accuracy of remote sensing at the Tibetan Plateau by comparing the results from the microwave remote sensing algorithm with the actual soil temperature measurements from 4 to 200 cm in those sites. They found the average difference between estimated and actual soil temperature was around 0.5 °C. Gómez et al. [51] used a regional climate model and remote sensing on soil temperature up to a 150 cm depth to simulate extreme heat events in Valencia, Spain. Given that some of the soil observation sites in Canada began recording data as early as 1958 [2], it is believed that these records can be joined with remote sensing data to create a longer observation period. This is especially important since many soil observation sites in Canada, particularly north of 60°, were too short to be used to create reliable climate analysis. Satellite data is not available in earlier years, making in-situ data collection the only source for those years. The measurements taken from the ground can be used to extend a location's soil temperature regime backwards. Due to the soil observation program shutting down by 1995 at Kuujjuaq and by the early 2000s across most stations in Canada, remote sensing can extend the availability of soil temperature data to create longer records and allow more accurate projections based on additional soil temperature data.

Houle et al.'s study [18] in southern Quebec achieved a correlation coefficient of ≥0.96 with the observed mean monthly temperature in their model output. This study's CanESM2 model achieved similar performance, as it produced a correlation coefficient of 0.95 to 0.96 between monthly mean modelled and observed temperature at the 5 to 20 cm depth (Table 2). Several notable differences between Houle et al. [18] and the current study were identified. While both of the studies were conducted in Quebec, one major difference is the distance between the sites involved. This study site was over 1000 km north of their northernmost site. Another major difference was the lack of incorporating snow data into this study's model. Lack of snow parameter in this model introduced inaccuracy due to the difference between air and soil temperatures because of the insulating effect of snow cover [52]. According to the climate normals at Kuujjuaq, the 30-year average snow depth between 1971–2000 was 19 cm [41]. and it dropped to 17 cm in the 1981–2010 normals period [42]. This contrasted with a relatively stable snow depth at the closest 30-year normal station, Hemon, Quebec in Houle et al. [18]'s study. In both the 1971–2000 and 1981–2010 normals periods, the average snow depth at Hemon were 21 cm [41,42]. Jean [43] also agreed that increasing soil temperature was associated with a decrease in

average maximum snow depth at Kuujjuaq. The higher variability of snow depth in this study posed challenges due to the absence of snow parameterization in the model. This study's model accuracy would continue to improve as the snow depth at this location was decreasing. The lack of trees and organic layer at the Kuujjuaq weather station (Figure 2) may have also unintentionally improved the model performance as it did not account for this variable. This was explained by Zhang et al. [52] as they found that the response of air temperature warming on soil temperature was quicker at climate stations which had low vegetation, such as Kuujjuaq, than at densely forested locations. The zero-curtain effect is not captured by this model as the SDSM program did not factor in the phase change of water in the soil during the overall warming nor the freeze-up process at the beginning of each winter.

### 4.3. Uncertainties

4.3.1. Observation Uncertainties

One of the major uncertainties around soil temperature projections in northern Canada was the scarce number of soil temperature monitoring sites. Most of the soil monitoring sites were located in southern parts of Canada [2] and the existence of those sites was primarily for agriculture purposes. Currently, soil temperature continues to be measured at many of these agriculture sites on an hourly scale. The current data has finer resolution than the historical data used in this study, where morning soil temperature was considered to be more complete than the afternoon temperature and the morning temperature was used as a representative of the temperature for the entire day [2]. This presents an uncertainty over the recorded temperature at the site. The current measuring instrument is believed to be more accurate than the older versions. The current temperature probe, Campbell Scientific CSI 109-L has a tolerance of $\pm0.1$ °C at 25 °C and the accuracy drops as temperature deviates away from 25 °C. This instrument reports the temperature up to one-hundredth of the decimal place. The older Northern Electric Type 14B thermistors that were used in this study had an accuracy of $\pm$ 0.1 °C with unknown temperature range for accuracy and the thermistors reported the values up to one-tenth of the decimal place. Unfortunately, the accuracy of soil data dropped when an installation of instruments in December 1978 [53] resulted in the recorded value to be rounded to the nearest 0.5 °C at all depths until the end of the study period in 1995. However, Jean [43] noted that there were no significant changes in observation or site methods for soil measurements at Kuujjuaq. It was unclear what caused a decrease in the observation accuracy and resolution.

Repairs to the thermistors would disturb the soil as the process involved digging the soil to retrieve the instrument from the ground and infill the soil after repairs were completed. Digging into the ground in northern sites also posed a challenge as it was common to encounter permafrost or bedrock before reaching the desired depth. Bedrock prevented Kuujjuaq from having soil temperature measurements at 300 cm [19]. Vegetation growth on top of the soil reduced the solar radiation from reaching the ground. Vegetation also reduced the evaporation rate by convection as it slowed the wind speed at the top surface of the soil. Majorowicz and Minea [54] suggested that the heat flow at Kuujjuaq was between 44 to 49 mW/m$^2$. However, the site conditions were likely compacted soil or gravel as the instrument was in the vicinity of the airport. Nonetheless, it was believed that the thermal diffusivity at Kuujjuaq was much higher than forested locations with moss or leaf litter as the organic layers in the forest were better insulators.

Soil moisture was not one of the variables measured at this site, but it was directly related to soil temperature and evaporation [55]. The coupled relationship between soil moisture and temperature would allow monitoring indirectly from satellite and extend the data availability beyond 1995. A dataset with a longer study period would improve the validation process and more robust in the accuracy of future projections. Since Kuujjuaq has an operational surface weather station up to present, soil moisture level can be derived via projected precipitation data [56]. However, as Beltrami and Kellman [17] pointed out, the coupling between air and soil temperature in Canada was site-specific. Having direct

soil temperature measurements on site would always be superior to drawing inferences from models through soil moisture and temperature coupling [55] or Stefan's equation to calculate the frost factor [12].

### 4.3.2. Modelling and Projections Uncertainties

The CanESM2 model was an improved model over older generations of Canadian GCMs. Chylek et al. [33] found that CanESM2 was more in agreement with observed air temperatures in the Arctic from 1900 to 1970 but it overestimated the warming after 1970. This raised a question on whether the warm bias would persist into the future. CanESM2 model incorporates the Canadian Land Surface Scheme (CLASS) which uses three depth layers at 0–10 cm, 10–35 cm and 35–410 cm [34]. Slater and Lawrence [57] described the third CLASS layer which was subjected to large temperature fluctuations because this layer has a shallow upper bound while the lower bound has low latent heat sink due to bedrock. This could explain some of the issues seen in winter where there was a cold bias for the model at 5 cm and a warm bias for depths at 10 cm and deeper. However, deficiencies in the land-surface model were not limited to CanESM2. As an example, HadGEM2 and GFDL's land models minimized the insulating effects of snow and caused cold bias in the soil [58,59]. On top of that, CanESM5 improved the radiative transfer for albedos with bare soil and snow in CanESM2 [37]. As shown in Figure 3, the CanESM2 model accurately predicted soil temperatures compared to the observed values when the monthly soil temperatures were greater than 0 °C (May to October). While improving snow albedo would generally improve the model accuracy, it had little effect on soil temperature since snow is a strong insulator and high snowfall amount at the study site [40–42] buffered some of the temperature variations in the soil [60]. On the other hand, Yokohata et al. [61] improved a component used in MIROC model to consider the heat capacity and thermal conductivity of frozen soil, organic layer near the surface and availability of unfrozen water with ground temperature below 0 °C. They found that MIROC model previously overestimated permafrost reduction area by about 15%. This demonstrated that both hot and cold biases could stem from a variety of reasons in different models but ultimately affected the projection of soil temperatures which would impact the forecast of permafrost distribution.

Under the present-day conditions, not using snow depth in the SDSM process introduced uncertainty into the projections. As the snow depth diminishes in the future, the uncertainties related to snow depth will decrease over time and can result in greater temperature fluctuations in deeper soil. The SDSM approach also failed to reproduce the general effects of zero-curtain effects or thermal lag in the deeper soil in both the historical and future time periods since the method was designed for climate projections on surface weather and not for soil. All of the SDSM predictors were above-surface variables, which introduced additional uncertainties for projecting below-surface parameters. Air temperature and surface precipitation were not prone to zero-curtain effects or thermal lag but soil temperature would often be affected by both factors. The zero-curtain effect is especially problematic to measure at this site due to the coarse resolution of the measuring instrument, especially after December 1978, as described in the previous section. A mathematical model may be required to account for zero-curtain effects on soil temperature. It should be noted that the CLASS-STEM modelling framework was recently transitioned to Canadian Land Surface Scheme including Biogeochemical Cycles (CLASSIC) model to improve usability and future model development [62].

There was a major difference between this study and some of the permafrost studies published in the 1990s in this region. In Wang and Allard [63] and Allard et al. [64], these studies used Kuujjuaq and Iqaluit's air temperature records to analyze permafrost distributions and soil temperatures at Salluit, Kangiqsujuaq and Quaqtaq which are located northwest of Kuujjuaq. Both studies suggested a recent cooling of air and soil temperatures in this area. Wang and Allard [63] observed a cooling rate in air temperature at 0.02 °C per year over a 40-year period from 1947 to 1988 in Kuujjuaq. Allard et al. [64] found that the

soil was cooling at 0.05 °C per year at the three sites and the cooling trend was observed at first 5 to 7 m of soil at Salluit. Using the observed air and soil cooling trends as the premise, Wang and Allard [63] modelled the future soil temperatures to 2044 at Salluit and found that the active layer thickness would decrease due to the continuing cooling trends at the surface. The observed trends in Wang and Allard [63] and Allard et al. [64] contrast greatly with more recent research in the same area such as this study, Boucher and Guimond [5] and Doré et al. [65]. This study found significant warming at the top 1.5 m of the soil (Table 1). Contrary to Wang and Allard [63] and Allard et al. [64], more recent studies by Fortier et al. [66], Boucher and Guimond [5] and Doré et al. [65] documented photographic evidence of permafrost degradation that was causing damage to the service road and drainage systems at the airports. Unlike Wang and Allard [63] which suggested surface cooling in the future, this study found that warming would continue in the future at all the depths (Figures 9 and 10). Wang and Allard [63]'s results also contrasted with the CanESM2's GCM output for soil temperature [35]. Therefore, Wang and Allard [63] and Allard et al. [64] drew different conclusions in their studies based on the trends in the time period that they studied and newer research appeared to have invalidated their findings.

The CanESM2, without SDSM, produced historical soil temperature at this site poorly. Using January 1967's monthly mean soil temperature as an example, the observed temperature was −23.4 °C at 5 cm and −22.3 °C at 10 cm. Without SDSM, CanESM2's historical soil temperature at first depth layer (0–10 cm) within Kuujjuaq's gridded cell was −7.9 °C while the downscaled 20-ensemble mean was −15.2 °C for 5 cm and 10 cm was −14.8 °C. The results demonstrated that SDSM vastly improved the historical CanESM2 estimation of soil temperature at this location. Further analysis showed that sometimes CanESM2 gridded data were slightly more accurate than SDSM in the summer. In August 1995, the monthly observed mean soil temperature was 12.0 °C at 5 cm and 11.2 °C at 10 cm. The gridded data estimated the temperature to be 11.9 °C between 0 and 10 cm while SDSM's 20-ensemble mean was 10.8 °C at 5 cm and 10.0 °C at 10 cm. However, in the 2 months prior (June 1995), the observed soil was 10.3 °C at 5 cm and 8.3 °C at 10 cm while the SDSM modelled 8.5 °C at 5 cm and 6.9 °C at 10 cm. Both of the SDSM modelled values were better than gridded data's temperature, which was 0.9 °C.

With the expected warming in the region, organic matter and vegetation such as moss would alter the surface layers of the thermal conductivity and hydraulic properties of the soil and this needed to be captured by models [67]. The decomposition rate of organic matter in soil was projected to increase in the future, partly caused by the increase in soil temperature [68]. However, most Earth system models perform poorly in permafrost regions due to the lack of permafrost carbon dynamics in the current models [68]. Weather station inspection report in 1972 confirmed coniferous trees were growing in the surrounding area of Kuujjuaq [20], indicating that the site was favourable for grass and shrubs to grow at the surface of the soil where the measurement took place. However, existing site management practice dictated active trimming of grass and removal of shrubs growing on top of the measuring site [23]. As the temperature warms at Kuujjuaq and the climate allows more vegetation to grow in the surrounding areas, this raised a question whether the soil temperature records located on a barren surface will be truly representative of the surrounding areas with trees and shrubs. In addition, Burke et al. [67] highlighted the importance of a model in capturing the feedback cycle of additional $CO_2$ emission through melting permafrost because the effects would also affect the salinity and humidity of the soil, with the change in the latter variable strongly linked to changes in soil temperature [55].

*4.4. Implications*

4.4.1. Implications to the Environment

As mentioned by Anderson et al. [7], there is an intricate relationship between soil, water and the climate, which in turn affect the hydrological cycle, local ecology and wildlife. With the projected warming of both air and soil temperatures in the area, this would lead to

drier soil [46]. Lower soil moisture could cause the soil to be more susceptible to drought conditions because the water, in liquid form, could evaporate more easily than ice [69]. Thawing of the permafrost would lead to higher decomposition rate [68] of existing organic materials that were previously frozen in ice. The decomposition process would lead to additional greenhouse gases being released into the atmosphere, further contributing to a positive feedback cycle. Indirectly, the melting permafrost would lead to more methylmercury being released into the aquatic ecosystem as the released carbon would provide suitable conditions for microbes to convert inorganic mercury into methylmercury [70]. The bioaccumulation and magnification of methylmercury would present challenges for aquatic animals and humans in the Arctic region. It is suggested that small changes in the local environmental factors could lead to a substantial response in the permafrost [43]. Warmer soil temperature would lead to higher vegetation productivity in the Arctic tundra environment [8].

### 4.4.2. Implications to Airports

Common impacts of degrading permafrost to airport infrastructure include disruption to the drainage system and the thaw settlement of the runways, taxiways and access roads [5,71]. At the nearby Tasiujaq airport, a study was conducted to assess the effectiveness of various methods that enhanced surface cooling or heat removal to lower the impacts of permafrost degradation on the runway [65]. They found that snow removal and heat drains with pipes installed at the shoulders of the runway were cost-effective measurements to preserve the permafrost underneath the runway. However, rising air temperature and diminishing snow depth at Kuujjuaq [40–42] may have unintended consequences of increasing rare occurrences of frostquakes, which causes popping sounds and cracks on the ground if the ground is saturated with water [72]. Warmer temperatures led to days with temperature above 0°C more often particularly in the fall and spring and thinner snow cover lessen the moderating temperature effects on soil. Both factors were crucial for frostquake because it required large temperature swings to occur. Active removal of snow cover in an attempt to allow rapid subsurface cooling to preserve permafrost could lead to even greater fluctuation in soil temperature and more prone to formation of frostquakes. However, it is believed that enhanced surface cooling of the runway will more than offset the drawbacks from cracks caused by frostquakes as permafrost would damage the airport infrastructure on a wider scale. Another study conducted at Iqaluit Airport in Nunavut also stated that surface snow cover as the greatest factor in the permafrost thaw and snow-covered edges along the taxiways and access roads would be most vulnerable to thaw settlement [73]. One of those taxiways was moved after multiple repairs failed to rectify the permafrost subsidence [74]. The Nunavik region (including Kuujjuaq and Tasiujaq) communities were characterized as low to moderately susceptible to hazards posed by climate change [75]. Given that Kuujjuaq is relatively close to Tasiujaq, the challenges and mitigation solutions to the runway integrity at Kuujjuaq are expected to be similar to Tasiujaq. Aside from the deterioration of the permafrost, airports in this region had been facing additional risk factors, such as higher wind speeds, that could further damage the infrastructure [76]. As a result, aviation infrastructure at Kuujjuaq is expected to face higher maintenance cost and more frequent inspections and repairs. On top of that, mitigation strategies will involve additional costs. Climate change projections in five Canadian airports with permafrost presence suggested that permafrost degradation affecting airport infrastructure could be more widespread and posed transportation challenges in wider regions across northern Canada [12,77]. This should not be a surprise for policymakers in the government, as they had considered the load-bearing capacity of permafrost underneath the airport as early as in the mid-1950s [78] and ground stability problems due to climate change in the late 1980s [79]. However, there was no comprehensive plan to investigate all northern airports in Canada to assess this known vulnerability.

Infrastructure resilience and responsible sourcing of construction materials for the climate change impacts on airports are two of the 17 Sustainable Development Goals (SDG)

outlined by the United Nations [80,81]. Responding to these challenges is a slow and challenging process due to the remoteness of these communities. For example, Iqaluit Airport's construction materials had to be carefully planned and ordered in advance to take advantage of lower shipping costs from summer sealift over delivering them by air [74]. Kuujjuaq and many villages in Nunavik have sea access through Hudson Bay and Hudson Strait, which makes them ideal candidates to ship supplies by summer sealift to address climate change impacts and implement adaptation strategies as occurrences of fog and low visibility decreased [82]. This reduces the communities' reliance on shipping freights by air and extends the shipping season in the summer.

*4.5. Long-Term Monitoring Network*

This research would not be possible without the existence of long-term soil temperature stations in a cold environment. Climate research relies on long-term observation records to accurately establish historical baselines on which the future projections are based. The consequences of short-term government priorities such as budget cuts [21] on environmental monitoring programs might not be felt until years or decades later. The soil monitoring at Kuujjuaq might not appear to be useful at the time when the program review determined it to be low impact or significance, but this research demonstrated its usefulness in the present day when Arctic warming becomes a global concern. Historical data in the Arctic are scarce, highlighting the usefulness of sites such as Kuujjuaq that provided a baseline for comparison with recent data. Lack of long-term monitoring led by the government cannot be offset by ad-hoc weather stations or boreholes records such as those at Salluit, Kangiqsujuaq and Quaqtaq northwest of Kuujjuaq [64] due to agency needs, political priorities at the time or university research projects as the sites are not intended for long-term data collection in this region [43] and are primarily tied to the funding cycle. Furthermore, the upkeep cost for in-situ soil sensing is expensive [7]. Even though these ad-hoc stations tended to be maintained, the sensors were often not calibrated [83] and thus the data quality from these stations was uncertain and questionable. Moreover, government weather stations had the advantage of having longer time series and met the standards set out by the World Meteorological Organization [43] which allow data to be exchanged globally and used in the meteorological forecast, research and GCMs. Ad-hoc weather stations could supplement but not replace the data collection done by government weather stations. Therefore, consistent and stable funding to various monitoring networks is needed for reliable atmospheric and geophysical data collection.

**5. Conclusions**

This study used in-situ, long-term soil temperature records at Kuujjuaq and found significant warmings of 0.9 to 1.3 °C per decade on an annual scale and 0.8 to 1.9 °C in winter at 5 to 150 cm depths. Soil at deeper depths had a lesser day-to-day variability and warmed more slowly than the surface soils. SDSM modelled the mean and variance of historical soil temperature data accurately, particularly from April to October and in soils closer towards the surface. Surface specific humidity, mean 2 m air temperature, 500 hPa geopotential height and specific humidity at 850 hPa showed the strongest correlation to Kuujjuaq's historical soil temperature. SDSM approach produced results that are close to the observed data and are robust as shown by the very small model errors. By 2026, soil temperature projections suggested that the annual mean will be above 0 °C in the three RCP scenarios. Winter mean soil temperature from November to April would remain below 0 °C. Soil temperatures would continue to rise by 2056 and 2086. The warming rate was linked to the greenhouse gas emission levels. Implications of the warming include possible cracks forming on the runway, damaging fuel tanks or compromising the structural integrity of terminal buildings that support the operation of the airport due to the degrading permafrost and freeze-thaw events each year.

Future directions of this study would involve improving the accuracy of winter projections. This could be achieved by performing another model validation process

exclusively for winter months (November to April) and identifying atmospheric variables which were best correlated to soil temperature in winter or a specific month to remove the winter bias in the model. Projections should also be attempted by using other climate models or with an ensemble approach. Finally, the SDSM process should be applied at another location with long-term soil temperature records to ensure that the method is applicable elsewhere.

This study established a strong linkage between soil temperature and CanESM2 model output through statistical downscaling. It also demonstrated that a novel usage of SDSM software was applicable to future soil temperature projections. This approach showed that using the SDSM approach on the CanESM2 model, the downscaled historical soil temperature was more accurate than the CanESM2 output for historical soil temperature at this location. The tool performs reasonably well in summer as well as in soil up to 100 cm deep. Through this study, it illustrated the importance of long-term government-funded supplementary network data such as soil temperature to monitor climate change in the north.

**Author Contributions:** A.C.W.L. conceptualized the experiment, methodology and procurement of data; A.C.W.L. and T.M. performed software model parameterization and validation; A.C.W.L., W.A.G. and T.M. conducted statistical analysis and visualization of the results; A.C.W.L., W.A.G. and T.M. prepared original draft, editing, revision and supervision of the project; W.A.G. acquired funding and resources of this project. All authors have read and agreed to the published version of the manuscript.

**Funding:** This research was funded by Natural Sciences and Engineering Research Council of Canada (NSERC), grant number RGPIN-2018-06801.

**Institutional Review Board Statement:** Not applicable.

**Informed Consent Statement:** Not applicable.

**Data Availability Statement:** The data presented in this study are available from https://climate.weather.gc.ca/ (accessed on 4 January 2022).

**Acknowledgments:** The authors would like to thank Kinson Leung from University of Toronto Scarborough for his assistance in the statistical downscaling in the SDSM software. A.C.W.L. would like to thank Kenneth Devine for providing information on soil measurement methods, site characteristics and photos through the station inspection reports. A.C.W.L. would also like to thank David Halliwell from Environment and Climate Change Canada for identifying the linkages between soil temperature and the atmospheric variables in the SDSM software.

**Conflicts of Interest:** The authors declare no conflict of interest.

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
