# Peer review of "Analysing Historical and Modelling Future Soil Temperature at Kuujjuaq, Quebec (Canada): Implications on Aviation Infrastructure"

_forecasting, doi:10.3390/forecast4010006_

Round 1

Reviewer 1 Report

The paper “Analysing Historical and Modelling Future Soil Temperature 2 at Kuujjuaq, Quebec (Canada): Aviation implications” describes the impact of climate change on soil temperatures and its effect on the aviation infrastructure due to permafrost thawing.

The paper employs SDSM in combination with CMIP5 model data to project soil temperatures.

In general, the paper reads well, and it is well structured giving a comprehensive

introduction about the topic and scopes for the work.

The main concern about the study is that relays on GCM data from AR5. Why haven’t the authors used the CanESM5 model instead (Swart, Neil C., et al. "The Canadian earth system model version 5 (CanESM5. 0.3)." Geoscientific Model Development 12.11 (2019): 4823-4873.)? Why have the authors chosen to use an older model? The use of the newer model could imply different outcomes on the analysis result. A clear motivation and justification for this would be in place in the paper.

If using CanESM5 instead of CanESM2 would not add substantial change in the result of the study, then this should be explicitly stated in the study.

Overall the methodology is consistent so I suggest that the paper would be conditionally accepted subject to the modifications or alternatively justifications above.

Citations missing:

Burke, Eleanor J., Yu Zhang, and Gerhard Krinner. "Evaluating permafrost physics in the Coupled Model Intercomparison Project 6 (CMIP6) models and their sensitivity to climate change." The Cryosphere 14.9 (2020): 3155-3174.

Melton, Joe R., et al. "CLASSIC v1. 0: the open-source community successor to the Canadian Land Surface Scheme (CLASS) and the Canadian Terrestrial Ecosystem Model (CTEM)–Part 1: Model framework and site-level performance." Geoscientific Model Development 13.6 (2020): 2825-2850.

Bourdeau‐Goulet, Sarah‐Claude, and Elmira Hassanzadeh. "Comparisons Between CMIP5 and CMIP6 Models: Simulations of Climate Indices Influencing Food Security, Infrastructure Resilience, and Human Health in Canada." Earth's Future 9.5 (2021): e2021EF001995.

Farquharson, Louise M., et al. "Climate change drives widespread and rapid thermokarst development in very cold permafrost in the Canadian High Arctic." Geophysical Research Letters 46.12 (2019): 6681-6689.

Yin, Guo-An, et al. "Data-driven spatiotemporal projections of shallow permafrost based on CMIP6 across the Qinghai‒Tibet Plateau at 1 km2 scale." Advances in Climate Change Research (2021).

Yokohata, Tokuta, et al. "Model improvement and future projection of permafrost processes in a global land surface model." Progress in Earth and Planetary Science 7.1 (2020): 1-12.

Minor comments:

Figure 1: the map could be zoomed out so that Labrador peninsula is more visible, to let the reader easily locate Kuujjuaq in the Canadian territory

On line 562 the second citation could be removed

Reviewer 2 Report

  1. The abstract is missing important results and especially policy implications. The first part of the abstract is too long and needs to be condensed. The authors need to learn to effectively and precisely communicate their findings and results. For example, you don't need to include everything in the abstract. Please try to focus on the most important information that you want to deliver to the audience.
  2. The introduction is not written well, and it does not flow the importance of the study to bring clarity of expression and bring across the contribution which in its current form is not obvious to the reader. Yet there is no mention that how this study makes a unique contribution to the current literature in the area. Yet there is no mention that how this study makes a unique contribution to the current literature in the area. Spell out upfront in the introduction section to highlight the significance of the study.
  3. Many abbreviations are missing such as NCEP/NCAR etc, and a few are did not used in the first place, such as NCEP, RCP, etc. Recheck the whole draft in terms of abbreviations used.
  4. Line 139, “Surface air temperature at 2 m from the same time period” what does ‘m’ mean here? Similarly at line 315, ‘850 hPa’. The unit of analysis for each variable should be mentioned clearly in the first place.
  5. It is better to use the notation for Pearson correlation (r) instead of Pearson r correlation and similarly for Pearson r coefficient.
  6. On lines 591-593 “This study’s 591 CanESM2 model performed better, as it achieved a correlation coefficient of >0.99 between 592 monthly mean modelled and observed temperaturebut the actual results did not show the correlation coefficient >0.99. All final results should be interpreted and compared accordingly.
  7. Nothing to mention regarding results robustness check and sensitivity analysis in the paper, it is better to run some methods to ensure the scientific soundness and the replicability of their study. There is no discussion about the accuracy of the simulated models such as the author(s) can use the accuracy of the model simulation by three indicators: the mean error (ME), the root mean squared error (RMSE), and the mean absolute relative error (MARE), etc.
  8. Data and study limitations should be listed which will way forward for future research.
  9. The conclusions section would benefit from the simplification and mainstreaming of the main findings. A wrap-up of the main achievements is also important which is not reflected in this case. It is a bit lengthy, make it short and to the point.
  10. Very weak and unconnected policy implications provided, it should be concrete according to the empirical results and few are general. Especially for the Aviation implications, many other factors can affect the aviation system such as COVID -19, etc. A clear link is missing between implications for aviation and the study results. Even it will be better to exclude the words “Aviation implications” from the title.

Reviewer 3 Report

Specific comments:

  • Line 80/81: why not using also Clyde River and/or Resolute data ?
  • Line 200: spatial resolution of CanESM2 ?
  • Line 208: the regional model was not used ? Spatial resolution ?
  • Line 240: how did you get 20 sim ? Perturbations ? Please explain
  • Line 262: the snow depth assumption is the most critical in my opinion, is there a way to take it into accont ?
  • Table 3: the first Pearson correlation should be surface specific humidity, not 500 hPa geopotential height
  • Line 356: 0°C instead of 0°C2
  • Figures 9 and 10: bad quality, please change them
  • Line 521: please refer to Fig 9 for annual temperature because Fig 10 refers to winter temperature

General comments:

  • according to the title, a closer look at the airports was expected, but there is only a small paragraph at the end of the article. I would strongly suggest to change the title giving less emphasis to airport problematic because it is not the core of the work
  • the usage of a global model instead of a regional model is highly impacting (in negative) and has many implications. For instance, in the model, are the land surface properties of the grid cell compatible with the Kuujjuaq ones ?

Round 2

Reviewer 1 Report

The authors addressed my comments and suggestions.

Reviewer 2 Report

Satisfied with the revised version